# ADAPTIVE BILEVEL OPTIMIZATION

## ABSTRACT

We propose a new *adaptive* optimization algorithm based on mirror descent for a class of possibly non-convex smooth bilevel optimization problems. The bilevel optimization template is broadly applicable in machine learning as it features two coupled problems where the optimal solution set of an inner problem serves as a constraint set for the outer problem. Currently, available algorithms require knowledge of both inner and outer gradient Lipschitz constants, which are difficult to tune in practice. By using an "on the fly" accumulation strategy on gradient norms, our adaptive algorithm circumvents this difficulty, and to our knowledge, is the first adaptive algorithm for bilevel optimization. In the convex setting, we obtain a convergence rate of $\mathcal{O}(1/T)$ in terms of the outer objective function, where $T$ is the number of iterations. When the outer objective is non-convex, our algorithm obtains an $\mathcal{O}(1/T)$ best-iterate guarantee for the squared norm of the gradient of the outer objective function. We also provide numerical evidence to support the theory in a reinforcement learning setting where all problem parameters are accessible.

## 1 INTRODUCTION

In this paper, we focus on bilevel minimization problems of the following form:

$$\begin{aligned} \min_{x \in \mathcal{X}} \quad & f(x; y^*(x)), \\ \text{such that} \quad & y^*(x) = \arg\min_{y \in \mathbb{R}^d} g(x; y), \end{aligned} \tag{BiOpt}$$

where $\mathcal{X}$ is a convex and a closed subset of $\mathbb{R}^d$. In the sequel, we assume that both the outer and inner objective functions $f$ and $g$ are continuously differentiable. We also assume that the solution set of the bilevel problem, which will be denoted by $\mathcal{X}^*$, is a non-empty subset of $\mathcal{X}$.

The bilevel problem (BiOpt) has found applications in many fields, most conventionally in game theory and more recently in machine learning (ML). In game theory, Stackelberg games (Von Stackelberg & Von, 1952) is the most prominent while, in ML, this template is useful in hyperparameter optimization (Shaban et al., 2019; Franceschi et al., 2017; 2018) and meta-learning (Rajeswaran et al., 2019; Franceschi et al., 2018). A detailed review of (BiOpt) applications can be found in (Colson et al., 2005; Dempe & Zemkoho, 2020; Sinha et al., 2017; Liu et al., 2021).

To obtain numerical solutions to (BiOpt), the "classical" approaches involve first-order methods (Shaban et al., 2019; Ghadimi & Wang, 2018; Franceschi et al., 2017), penalty methods (Lin et al., 2014), KKT reformulations-based approaches (Allende & Still, 2013), value function-based methods (Ye & Zhu, 1995; Sow et al., 2022), and trust-region algorithms (Marcotte et al., 2001). A common requirement in existing algorithms is the assumption of the knowledge of gradient Lipschitz constant for the inner and a "hyper" Lipschitz constant for the outer levels that depends in a complicated manner on many regularity parameters of both levels. Unfortunately, these constants are often unknown and are challenging to tune in practice.

To address the key issue in deploying these algorithms in practice, it is natural to ask the following:

*Is it possible to design a method for* (BiOpt) *without knowing the Lipschitz constants?*

In what will follow, we answer this question affirmatively based on the notion of *adaptivity*.

**Related work on bilevel optimization**  An increasingly large number of challenging applications can be formulated in (BiOpt), which has fueled the research on bilevel optimization algorithms. Progress in this setting mainly focuses on unconstrained inner-level problems under the assumption that the inner objective function is strongly convex with respect to the optimization variable of the inner level, for instance, see (Ghadimi & Wang, 2018; Hong et al., 2020; Khanduri et al., 2021a;b; Chen et al., 2021a; Ji et al., 2021; Chen et al., 2021b; Yang et al., 2021), for both stochastic and deterministic cases.

These structural assumptions are widely used in the setting of bilevel optimization problems, due to the fact that in this case, we can guarantee the existence of a gradient of the inner objective function and that it is relatively easy to compute it implicitly. In addition, the implicit gradient in these settings can be shown to be Lipschitz continuous under mild assumptions, for example, see (Ghadimi & Wang, 2018, Lemma 2.2) and (Khanduri et al., 2021b, Lemma 3.1). Following (Ghadimi & Wang, 2018), this paper also focuses on bilevel optimization problems with a continuously differentiable outer objective function, which has a Lipschitz continuous gradient.

The Lipschitz continuous gradient assumption is useful for the application of many optimization algorithms and also for obtaining their convergence results. However, when dealing with bilevel optimization, we argue that this assumption is even more crucial: Designing efficient iterative solution methods usually requires not only prior knowledge of the Lipschitz modulus but a so-called "hyper" Lipschitz constant, which depends on various parameters of both the layers of (BiOpt). This fact unfortunately can be proved to be quite a significant bottleneck from a computational point of view.

**Contributions.**  Merging the two realms of adaptivity and bilevel problems is not an easy task. In particular, the two-level nature of bilevel optimization does not allow to straightforwardly extending the standard arguments of adaptive methods initially aiming at the regular minimization framework. The main difficulty stems from the fact that the objective functions of outer and inner problems may be completely antithetical in terms of structural assumptions (for example, we explore problems where the outer objective is possibly non-convex meanwhile the inner objective is strongly convex). We next summarize the main contributions of the paper.

- Designing a novel adaptive method for smooth bilevel optimization problems. In particular, our method automatically adapts to both levels of a smooth bilevel problem without any prior knowledge of the respective Lipschitz constants or a fortiori of any of the more complicated hyper-constants mentioned above. To the best of our knowledge, our algorithm is the first adaptive method addressing (BiOpt).

- Our method achieves $\mathcal{O}(1/T)$ convergence rate guarantees, relative to the sub-optimality gap whenever the outer objective is convex and $\mathcal{O}(1/T)$ for the so-called best iterate for non-convex outer objectives.

- Providing an asymptotic convergence rate result for the last iterate of our method (i.e., the actual sequence without any averaging) towards the solution set. Finally, we establish a stronger convergence result towards a particular solution point under the assumption that the feasible domain is bounded.

## 2  PROBLEM SETUP AND PRELIMINARIES

The problem (BiOpt) consists of two levels of optimization problems. First, the inner level, which consists of an unconstrained minimization

$$\min_{y \in \mathbb{R}^d} g(x; y), \qquad\qquad \text{(Inner)}$$

with a minimizer $y^*(x)$ that affect the *outer optimization problem*

$$\min_{x \in \mathcal{X}} f(x; y^*(x)). \qquad\qquad \text{(Outer)}$$

This class of bilevel optimization problems, as discussed in the introduction, is broad enough to encompass a wide range of challenging applications. Another field that has seen significant presence of bilevel optimization problems and algorithms, is Reinforcement Learning (RL). We will discuss below, in detail, one particular setting that can be formulated as a bilevel optimization problem (see Appendix A).

**The oracle model.**  From an algorithmic point of view, we aim to solve (BiOpt) by using iterative methods that require access to *black-box oracles mechanisms* (Nesterov, 2004). Starting with the inner optimization problem, we assume that at each stage of the inner iterative process, the optimizer can query a *first-order* black-box mechanism, which returns the inner objective's (partial) gradient at the queried point. Formally, when the oracle mechanism is called at $y \in \mathbb{R}^d$, we assume that the gradient vector $\nabla_y g(x; y) \in \mathbb{R}^d$ is returned. In practice, the oracle will be called repeatedly at a sequence of points $y_t \in \mathbb{R}^d$ generated by the inner loop of the algorithm under study.

On the other hand, the above mechanism cannot be straightforwardly extended to the outer minimization process, since the computation of a gradient of the outer objective function requires the knowledge of $y^*(x)$, which in most cases is unknown. Therefore, we assume that the optimizer has access to the standard bilevel *hyper-gradient* oracle mechanism as defined in (Ghadimi & Wang, 2018), which serves in the sequel as an efficient surrogate for the unknown original gradient $\nabla f(x; y^*(x))$ as discussed in Lemma 2.2 of (Ghadimi & Wang, 2018). Formally, the optimizer has access to the following quantity:

$$\tilde{\nabla} f(x, y) = \nabla_x f(x, y) - M(x, y) \nabla_{x'} f(x, y), \qquad \text{(HyperGrad)}$$

where

$$M(x, y) = \nabla^2_{x,y} g(x, y) \left[ \nabla^2_{y,y} g(x, y) \right]^{-1}. \qquad (1)$$

Now, that we have at hand both black-box oracles for the computation/approximation of gradients of the inner and outer objective functions, in the following sections we will first discuss standard algorithms for solving (BiOpt) and their potential drawbacks, and then develop our adaptive algorithm.

## 3   THE APPROXIMATION BILEVEL METHOD

In this section, we first discuss the approximation bilevel method proposed in (Ghadimi & Wang, 2018)and finish with a discussion on the convergence rate guarantees which were achieved in this paper including the limitation of their method.

We begin with a short description of the approximation bilevel method. First, by exploiting the strong convexity and smoothness of the inner problem, it employs an $t_k$ inner loop of gradient descent steps, with respect to the gradient $\nabla_y g(x; y)$. More precisely, any inner iteration ($1 \le t \le t_k$) takes the following form:

$$y_{t+1} = y_t - \eta \nabla_y g(x_k; y_t), \qquad (2)$$

where $\eta > 0$ is a step-size.

Now, after the inner loop terminates, we have at hand an approximation $\bar{y}_k = y_{t_k}$ of the solution of the inner minimization problem. Moving to the outer level now, the optimizer applies a (projected) "hyper-gradient" step (see (HyperGrad)) with respect to the outer problem while fixing the inner variable fixed $\bar{y}_k$. Formally, the outer step is given by:

$$x_{k+1} = \arg\min_{x \in \mathcal{X}} \{ \gamma \langle \tilde{\nabla} f(x_k, \bar{y}_k), x - x_k \rangle + \|x_k - x\|^2 \}.$$

**Convergence guarantees of the approximation bilevel method.**  The theoretical results obtained in (Ghadimi & Wang, 2018), see also (Tarzanagh & Balzano, 2023) and references therein, are based on the following assumptions.

**Assumption 1.**  The outer objective function $f$ satisfies the following properties:

1. For any $x \in \mathcal{X}$, the gradients $\nabla_x f(x; y)$ and $\nabla_y f(x; y)$ are Lipschitz continuous with constants $L_x > 0$ and $L_y > 0$, respectively.

2. For any $x \in \mathcal{X}$ and $y \in \mathbb{R}^m$, $\|\nabla_y f(x; y)\|_* \le C_y$ with a constant $C_y > 0$.

3. For any $y \in \mathbb{R}^m$, $\nabla_y f(x; y)$ is Lipschitz continuous relative to $x$ with a constant $\bar{L}_y > 0$.

**Assumption 2.**  The inner objective function $g$ satisfies the following properties:

1. For any $x \in \mathcal{X}$ and $y \in \mathbb{R}^m$, $g$ is continuously twice differentiable in $(x, y)$.

2. For any $x \in \mathcal{X}$, $\nabla_y g(x, y)$ is Lipschitz continuous relative to $y$ with a constant $L_g > 0$.

3. For any $x \in \mathcal{X}$, $g(x, y)$ is strongly convex relative to $y$ with a modulus $H_g > 0$.

4. For any $x \in \mathcal{X}$, $\nabla^2_{xy}g(x,y)$ and $\nabla^2_{y,y}g(x,y)$ are Lipschitz continuous (w.r.t $y$) with constants $L_{g,x} > 0$ and $L_{g,y} > 0$.

5. For any $y \in \mathbb{R}^d$ $\nabla^2_{xy}g(x,y)$ and $\nabla^2_{y,y}g(x,y)$ are Lipschitz continuous (w.r.t $x$) with constants $\bar{L}_{g,x} > 0$ and $\bar{L}_{g,y} > 0$.

6. For any $x \in \mathcal{X}$, $y \in \mathbb{R}^d$, $\|\nabla^2_{xy}g(x,y)\| \leq C_g$ with $C_g > 0$.

Based on the above assumptions, (Ghadimi & Wang, 2018) derives the following rate of convergence of the approximation bilevel method:

$$f(x_T; y^*(x_T)) - f(x^*; y^*(x^*)) = \mathcal{O}(1/T),$$

where the outer objective function $f$ is convex. If $f$ is possibly non-convex the method guarantees:

$$\min_{1 \leq k \leq T} \|\nabla f(x_k; y^*(x_k))\|^2 = \mathcal{O}(1/T).$$

However, both results of the approximation bilevel method crucially rely on a particular choices of the involved parameters (as mentioned in the assumptions above) of the respective inner and outer objective functions. In particular, their theoretical analysis requires first to compute a generalized "Lipschitz" constant:

$$L_f = \frac{(\bar{L}_y + C)C_g}{H_g} + L_x + C_y \left[ \frac{\bar{L}_{g,x}C_y}{H_g} + \frac{\bar{L}_{g,y}C_g}{H_g^2} \right], \tag{3}$$

with the constant $C$ is given by:

$$C = L_x + \frac{L_y C_g}{H_g} + C_y \left[ \frac{L_{g,x}}{H_g} + \frac{L_{g,y}C_g}{H_g^2} \right] \tag{4}$$

and then set the two respective step-size policies as follows:

$$\eta = \frac{2}{L_g + H_g} \quad \text{and} \quad \gamma \leq 1/L_f \quad \text{for all} \ \ k \geq 0, t \geq 0.$$

A natural question which arises from the above step-sizes is whether these constants can be efficiently computed. First, in several practical regularized settings, where the strongly convex regularizing parameter is known by the user, one can safely observe that the strong convexity parameter can be easily determined.

On the other hand, obtaining the hyper-constant $L_f$ is computationally prohibitive, since it requires the Lipschitz constants of both the inner and outer objective functions. Note that, even if we have prior knowledge of the Lipschitz constants, the step-size is also determined by additional complicated expressions involving regularity conditions of both inner and outer problems. Similar difficulty, even though smaller, is present in the determination of the other step-size $\eta$.

As a next step, we will discuss the main shortcomings of such a dependency in more detail.

**Limitation of the approximation bilevel method.** As we mentioned above, the convergence rates of the approximation bilevel method do *not* hold, if the step-size is not suitably fine-tuned. In order to illustrate this in practice, we consider both levels of (BiOpt) to be quadratic functions. That is, $f, g : \mathbb{R}^2 \times \mathbb{R}^2 \to \mathbb{R}$ are defined as follow $g(x; y) = (100x_1 - 10)^2 + (x_2 - 2)^2$ and $f(x; y) = x^T \text{diag}(y)x$, where $x_i$ denotes the $i^{\text{th}}$ coordinate of the vector $x$. Since both levels are of strongly convex functions, the approximation bilevel method requires $\eta \leq 2/(H_g + L_g) = 2/20002 \approx 1e-4$ and $\gamma \leq 2/(H_f + L_f) = 0.95$. In Figure 1, we show that the approximation bilevel method diverges when ran with the maximum allowed value for $\eta$ and a slightly overestimated value for $\gamma$. Moreover, we show that the same behaviour happens when $\gamma$ is well tuned but the inner problem step-size $\eta$ is not. In stark contrast, our Algorithm 1 (to be presented shortly) converges reliably without requiring knowledge of neither $L_f$ or $L_g$.

As we also illustrate in Figure 1, in practice, small deviations from the theoretical step-sizes can cause catastrophic oscillations and even divergence. This phenomena also occurs in the standard minimization setting, which originally motivated the developments of adaptive methods in order

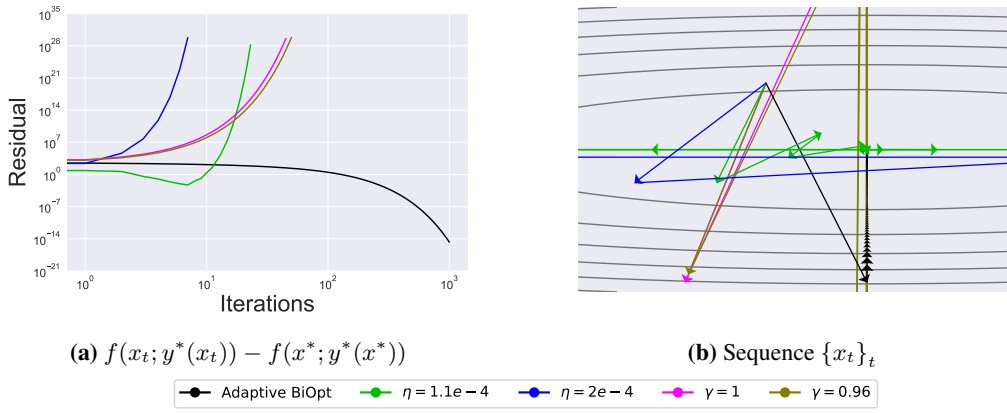

**(a)** $f(x_t; y^*(x_t)) - f(x^*; y^*(x^*))$        **(b)** Sequence $\{x_t\}_t$

Adaptive BiOpt    $\eta = 1.1e-4$    $\eta = 2e-4$    $\gamma = 1$    $\gamma = 0.96$

**Figure 1:** Quadratic (BiOpt) problem where our Algorithm 1 converges while the approximation method diverges if run with even just one misspecified step-size ($\gamma$ or $\eta$). In the legend, we write the value of the misspecified step-size while the other is set as required by the approximation method.

to handle these situations, see for instance, (Antonakopoulos et al., 2021; Nesterov, 2015; Kavis et al., 2019; Levy et al., 2018; Bach & Levy, 2019) and references therein. In the setting of bilevel optimization problems, such as (BiOpt), the situation becomes even more complicated, since the optimizer should *simultaneously* tune the step-sizes of both the inner and outer problems as detailed above in approximation bilevel method.

Additionally, note that this tuning can be quite demanding in real-world problems, since obtaining the knowledge of these constants is often impractical and computationally prohibitive.

## 4  ADAPTIVE BILEVEL MIRROR DESCENT

In this section, we aim to propose an adaptive method, which overcomes the above shortcomings by introducing adaptive step-size policies for both the inner and outer loops. Before discussing the main ideas behind our algorithm, we will need some mathematical notions that will be useful for the development of our algorithm.

**Definition 1.** We say that $h : \mathbb{R}^d \to \mathbb{R} \cup \{\infty\}$ is a *distance generating function* on $\mathcal{X}$ if:

1. $h$ is convex, lower semi-continuous (l.s.c.), $\mathrm{cl}(\mathrm{dom}\, h) = \mathrm{cl}(\mathcal{X})$, and $\mathrm{dom}\, \partial h = \mathcal{X}$.
2. The subdifferential of $h$ admits a *continuous selection* $\nabla h(x) \in \partial h(x)$ for all $x \in \mathcal{X}$.
3. $h$ is *strongly convex*, i.e., there exists some $K > 0$ such that

$$h(x') \geq h(x) + \langle \nabla h(x), x' - x \rangle + \tfrac{K}{2} \|x' - x\|^2, \quad \forall x \in \mathcal{X}, x' \in \mathrm{dom}\, h. \tag{5}$$

The *Fenchel coupling* induced by a regukarizer $h$ is then defined by

$$F(x, v) = h(x) + h^*(v) - \langle v, x \rangle, \tag{6}$$

where $h^*$ denotes the Fenchel conjugate of $h$. For a panoramic view of the above toolbox we refer the reader to (Chen & Teboulle, 1993) for examples of distance-generating functions.

First, we begin by presenting the adaptive inner loop. Namely, at an outer iteration counter $k$, we run $t_k$ inner iterations, which compute a (normalized) gradient descent step

$$y_{t+1} = y_t - \eta_t \frac{\nabla_y g(x_k; y_t)}{\|\nabla_y g(x_k; y_t)\|^2}, \tag{7}$$

where the step-size $\eta_t$ is computed using the following adaptive idea. Obviously, the agnostic behavior relative to the Lipschitz modulus of our method crucially relies on the choices of the adaptive step-sizes. Therefore, we devote the following paragraph to describe it in a detailed fashion.

**Adaptive step-size for the inner loop.** The adaptive step-size $\eta_t$ is mainly inspired by the generic optimization setup of (Levy, 2017). Formally, our adaptive step-size is defined, for any inner iteration

$t$, using two computations. First, we aggregate the squared norms of the gradients up to the current iteration counter $t$:

$$G_t = G_{t-1} + \|\nabla_y g(x_k; y_t)\|^2, \tag{8}$$

and then $\eta_t$ is given by $\eta_t = (H_g G_t)^{-1}$, recall that $H_g$ is the strong convexity parameter of $g$.

An intuition behind the success of the proposed adaptive step-size policy comes from the simple minimization setting. In particular, it is a standard result that for strongly convex objective functions an appropriate step-size should scale as $\eta_t \propto 1/t$ as opposed to $\eta_t \propto 1/L$ for the smooth and convex case. Therefore, as it is pointed out in (Levy, 2017), a reasonable mechanism for the automatic adjustment of the inner loop method is that $\eta_t$ interpolates between these to extreme learning rates. For the strongly convex case, one can observe that the step-size behaves asymptotically as $1/t$ if the gradients are of the same magnitude.

We continue with the description of our inner loop. After $t_k$ inner iterations, we generate the following output

$$\bar{y}_k = \left[ \sum_{s=1}^{t_k} \|\nabla_y g(x_k, y_s)\|^{-2} \right]^{-1} \sum_{s=1}^{t_k} \|\nabla_y g(x_k, y_s)\|^{-2} y_s. \tag{9}$$

Now, we are ready to discuss our modification of the outer loop. Using the output of the inner loop $\bar{y}_k$, the algorithm continues with the (lazy) mirror descent step:

$$U_{k+1} = U_k - \gamma_k \tilde{\nabla} f(x_k; \bar{y}_k), \quad x_{k+1} = Q(U_{k+1}) = \arg\min_{x \in \mathcal{X}} \{\langle U_{k+1}, x \rangle + h(x)\}. \tag{MD}$$

Note that the update steps given in (MD) can be seen as a generalization of the so-called lazy gradient descent. Due to space constraints, we omit a more detailed discussion and point interested readers to an overview (Shalev-Shwartz, 2011).

Also in this case we incorporated an adaptive step-size $\gamma_k$, which is discussed next.

**Adaptive step-size of the outer loop.** We begin with a discussion on how to define adaptive step-size in the unconstrained case. In this setting, a popular adaptive choice is the so-called "inverse-sum-of-squares" policy

$$\gamma_k = 1 \Big/ \sqrt{\sum_{s=1}^k \|\nabla f(x_s; y^*(x_s))\|^2}, \tag{10}$$

where $x_s$ is the iterates produced by the algorithm (Duchi et al., 2011; McMahan & Streeter, 2010). However, this choice of $\gamma_k$ does not seem to be useful in the setting of bilevel optimization. First, as we discussed earlier, the nested nature of (BiOpt) does not allow to access $\nabla f(x_s; y^*(x_s))$. Moreover, if the problem is constrained, the extra terms entering the denominator of $\gamma_k$ do not vanish as the algorithm approaches a solution, so the step-size given in (10) may still fail to exploit the objective function smoothness. Therefore, we propose the following modification for the adaptive step-size that fits the bilevel setting. Let us begin with the simplest Euclidean case. In this case, since the difference $\|x_{k+1} - x_k\|$ must always vanish near a solution (even near the boundary), we can use it as an approximation for $\nabla f(x; y^*(x))$ in constrained problems. This idea is formalized by the notion of *gradient mapping* (Nesterov, 2004), which for the simple minimization setting and can be defined here as

$$\delta_k = \|x_{k+1} - x_k\| \big/ \gamma_k. \tag{11}$$

On the other hand, in the mirror descent setting, the prox-mapping tends to deflate gradient steps, so the norm difference between two successive iterates $x^+$ and $x$ could be very small relative to the oracle signal that was used to generate the update. As a result, the Euclidean residual (11) could lead to a disproportionately large step-size that would be harmful to convergence. For this reason, we accumulate gradient mappings that take into account the Fenchel coupling geometry of the method. Mathematically speaking, we define

$$\delta_k = \sqrt{F(x_k, U_{k+1}) + F(x_{k+1}, U_k)} \big/ \gamma_k. \tag{12}$$

Obviously, when $h(x) = (1/2)\|x\|_2^2$, we readily recover the definition of the Euclidean gradient mapping (11). In general, however, by the strong convexity of $h$, the value of this "Fenchel residual" exceeds the corresponding Euclidean definition, so the induced step-size exhibits smoother variations that are more adapted to the bilevel framework. To summarize, we provide the complete pseudo-code of our algorithm

---

**Algorithm 1:** Adaptive Bilevel Optimization

---

1: **Input:** Initial points $x_0 \in \mathcal{X}$, $y_0 \in \mathbb{R}^d$, $\{t_k\}_{k \geq 0}$ integer sequence, $h$ regularizer, $H_g$ strong convexity parameter of $g$.

2: **for** $k = 1$ to $T$ **do**

3:    $G_0 = 0$

4:    **for** $t = 1$ to $t_k$ **do**

5:       Compute step-size $\eta_t = (H_g G_t)^{-1}$ with $G_t = G_{t-1} + \|\nabla_y g(x_k, y_t)\|^2$.

6:       Update $y_{t+1} = y_t - \eta_t \frac{\nabla_y g(x_k, y_t)}{\|\nabla_y g(x_k, y_t)\|^2}$.

7:    **end for**

8:    Compute approximate solution $\bar{y}_k = \left[ \sum_{s=1}^{t_k} \|\nabla_y g(x_k, y_s)\|^{-2} \right]^{-1} \sum_{s=1}^{t_k} \|\nabla_y g(x_k, y_s)\|^{-2} y_s$.

9:    Compute step-size $\gamma_k = \left[ 1 + \sum_{j=1}^{k-1} \delta_j^2 \right]^{-1/2}$ with $\delta_j^2 = \frac{F(x_j, U_{j+1}) + F(x_{j+1}, U_j)}{\gamma_j^2}$.

10:    Aggregate hypergradients $U_{k+1} = U_k - \gamma_k \tilde{\nabla} f(x_k; \bar{y}_k)$.

11:    Update $x_{k+1} = Q(U_{k+1})$.

12: **end for**

---

## 5   Results

We are now in a position to present our main convergence result for Algorithm 1. We start by providing the convergence rate guarantees of Algorithm 1 for convex and non-convex variants of (BiOpt). Formally, we have the following result.

**Theorem 1.** *Assume Assumption 1 and Assumption 2 hold, let $\{x_k\}_{k \geq 0}$ and $\{\bar{y}_k\}_{k \geq 0}$ be the iterates generated by Algorithm 1. We denote $\bar{x}_T = \sum_{k=1}^{T} x_k / T$. Then, by choosing $t_k = \lceil k^{1/4} \rceil$, the following estimations hold :*

*1. If $\mathcal{X}$ is bounded and $f$ is convex then,*

$$f(\bar{x}_T; y^*(\bar{x}_T)) - f(x^*; y^*(x^*)) = \mathcal{O}(1/T). \tag{13}$$

*2. If $f$ is non-convex and $\mathcal{X} = \mathbb{R}^m$ then,*

$$\min_{1 \leq k \leq T} \|\nabla f(x_k; y^*(x_k))\|^2 = \mathcal{O}(1/T). \tag{14}$$

Theorem 1 shows that Algorithm 1 recovers the same bounds as of the approximation bilevel method, while being agnostic to the Lipschitz constants of (BiOpt). The starting point of our proof (which we detail in the supplement) uses the following regret bound for the case of convex outer problems.

**Proposition 1.** *Using the notation of Theorem 1, and $f$ being convex. Then, by setting $\theta_k = f(x_k; y^*(x_k)) - f(x^*; y^*(x^*))$, Algorithm 1 enjoys the following regret bound, for all $x^* \in \mathcal{X}^*$:*

$$\sum_{k=1}^{T} \theta_k \leq \sum_{k=1}^{T} \left( \frac{1}{\gamma_{k+1}} - \frac{1}{\gamma_k} \right) F(x^*, U_{k+1}) + \operatorname{diam} \mathcal{X} \sum_{k=1}^{T} \sqrt{\frac{2e(1 + H_g/L_g)t_k}{H_g^2 e^{H_g/L_g t_k}}} + \sum_{k=1}^{T} \gamma_k \delta_k^2.$$

In particular, the "exponentially decaying" term arises from the inner loop adaptive approximation of the strongly convex and smooth inner minimization. The next step is to show that $\sup_{x^* \in \mathcal{X}^*} F(x^*, U_{k+1})$ is a bounded sequence. Finally, the key component for obtaining an $\mathcal{O}(1/T)$ rate is the stabilization of the outer step-size sequence and the fact that it has a strictly positive limit as we state next in Lemma 1. An important feature of this property is that it suggests the method does not slow down near a solution.

**Lemma 1.** *Under the notation of Theorem 1, the Fenchel residual sequence $\delta_k$ satisfying $\sum_k \delta_k^2 < +\infty$. Moreover, the outer step-size sequence $\{\gamma_k\}_{k \geq 0}$ converges to some $\gamma_\infty > 0$.*

The above stabilization result is actually obtained independently from the convex structure of the outer objective function, and therefore immediately implies the best-iterate convergence rate. We conclude this section with another result in the non-convex setting, which gives a last iterate convergence guarantee. More precisely, the trajectory of the generated iterates by our algorithm, satisfying the

"secant inequality" (Bottou, 1998; Mertikopoulos & Zhou, 2019; Mertikopoulos et al., 2019; Zhou et al., 2020b)

$$\inf\{\langle \nabla f(x; y^*(x)), x - x^* \rangle : x^* \in \arg\min f, x \in \mathcal{K}\} > 0, \tag{SI}$$

for every closed subset $\mathcal{K}$ of $\mathcal{X}$ that is separated by neighborhoods from $\arg\min f$. Variants of this condition have been widely studied in the literature and include non-convex functions with complicated ridge structures (Nevel'son & Khasminskii, 1976; Ljung, 1978; Bottou, 1998; Facchinei & Pang, 2003; Jiang & Xu, 2008; Zhang & Yin, 2013; Karimi et al., 2016; Zhou et al., 2017). It is easy to verify that (SI) always holds for convex functions. Formally, we have the following theorem.

**Theorem 2.** *Using the notation of Theorem 1, the following hold:*

1. *The sequence $\{\mathrm{dist}(x_k, \mathcal{X}^*)\}_{k \geq 0}$ converges to $0$.*

2. *If $\mathcal{X}$ is, in addition, compact, then $\{x_k\}_{k \geq 0}$ converges to some point $x^* \in \mathcal{X}^*$.*

The main idea of the proof (which we detail in the appendix) consists of two steps. First, we establish that fact $\liminf f(x_k) = \min_{x \in \mathcal{X}} f(x; y^*(x))$. Then, we conclude the analysis by invoking arguments that involve a quasi-Fejér argument as in (Combettes, 2001; Bottou, 1998). We delegate the technical arguments to the supplement.

## 6 NUMERICAL EXPERIMENTS

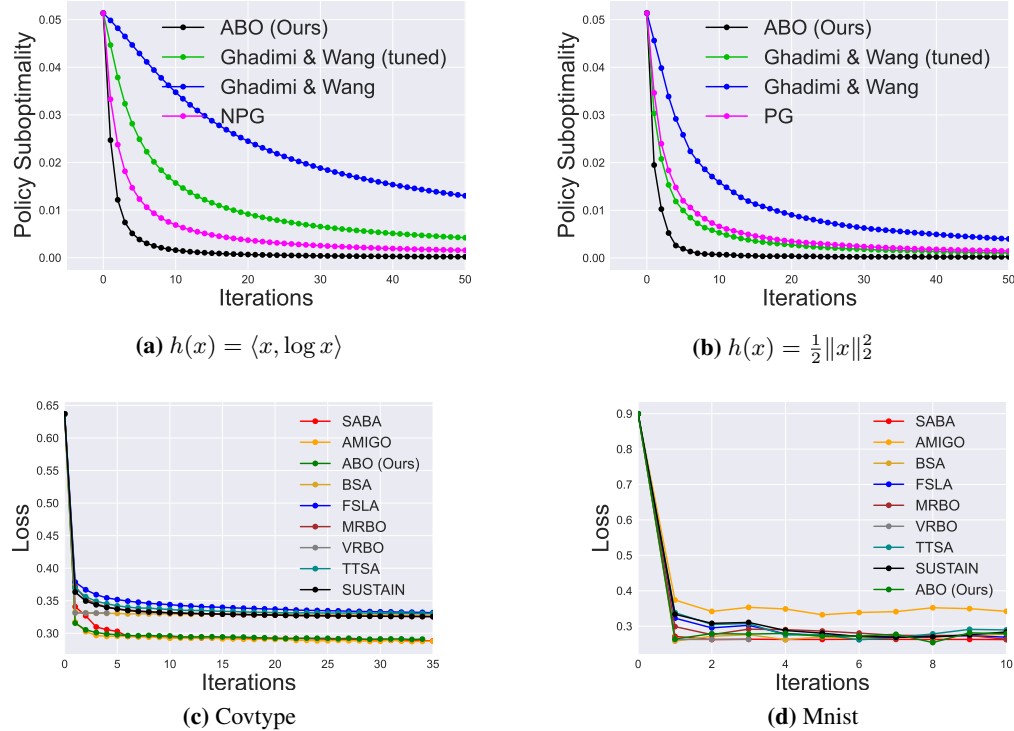

**(a)** $h(x) = \langle x, \log x \rangle$

**(b)** $h(x) = \frac{1}{2}\|x\|_2^2$

**(c)** Covtype

**(d)** Mnist

**Figure 2:** Figures 2a and 2b: Comparison between Ghadimi & Wang (2018) with our Algorithm 1 for solving (Actor-Critic). We also notice superior performance compared to (Natural) Policy Gradient (Agarwal et al., 2020; Sutton et al., 1999) abbreviated NPG and PG.
Figures 2c and 2d: Comparison with other recent non adaptive bilevel optimization method on the benchmarks available at https://github.com/benchopt/benchmark_bilevel.

Here, we demonstrate numerically how our adaptive method Algorithm 1 can be applied to an example from the domain of Reinforcement Learning, as described below. In RL, the environment and its underlying dynamics are typically abstracted as a Markov Decision Process (MDP) Puterman (1994) given by a tuple $M = (\mathcal{S}, \mathcal{A}, P, \boldsymbol{\nu}_0, c, \gamma)$. In this setting, $\mathcal{S}$ is the state space, $\mathcal{A}$ is the action space, $P : \mathcal{S} \times \mathcal{A} \to \Delta_{\mathcal{S}}$ is the transition law, $\boldsymbol{\nu}_0 \in \Delta_{\mathcal{S}}$ is the initial state distribution,

$c \in [0,1]^{|\mathcal{S}||\mathcal{A}|}$ is the one-stage cost, and $\gamma \in (0,1)$ is a discount factor. A *stationary Markov policy* is a mapping from states to action distributions, i.e. $\pi : \mathcal{S} \to \Delta_{\mathcal{A}}$ the set of policies is denoted as $\Pi$. For any policy $\pi$, we define the *state action value function as* $Q : \Pi \times \mathcal{S} \times \mathcal{A} \to \mathbb{R}$ as follows $Q(\pi, s, a) \triangleq \mathbb{E}_\pi \left[ \sum_{t=1}^\infty c(s_t, a_t) | s_1 = s, a_1 = s \right]$, that is the expectation of the *cumulative discounted cost* incurred controlling the MDP $M$ with policy $\pi$. In particular, the expectation is over the randomness of the trajectory $\{(s_t, a_t)\}_{t=1}^\infty$ induced by the transition law $P$ and by the policy $\pi$. The goal of Actor-Critic methods, is to solve the following optimization problem $\pi^\star \in \arg\min \mathbb{E}_{s \sim \boldsymbol{\nu}_0} \left[ \langle Q(\pi, s, \cdot), \pi(\cdot|s) \rangle \right]$. It is known that denoting $Q(\pi)$ to denote the vector with entries $Q(\pi, s, a)$ for some canonical ordering of $s, a$, In addition, using $Q(\pi)$ to denote the vector with entries $Q(\pi, s, a)$ for some canonical ordering of $s, a$, it is known that $\pi^\star$ can be found solving the following instance of (BiOpt).

$$\min_{\pi \in \Pi} \quad \mathbb{E}_{s \sim \boldsymbol{\nu}_0} \left[ \langle Q^*(\pi, s, \cdot), \pi(\cdot|s) \rangle \right],$$
$$\text{such that} \quad Q^*(\pi) = \underset{y \in \mathbb{R}^{|\mathcal{S}||\mathcal{A}|}}{\arg\min} \|c + \gamma P^\pi y - y\|^2. \qquad \text{(Actor-Critic)}$$

The above formulation led to the development of influential algorithms (Peters & Schaal, 2008; Schulman et al., 2015; 2017).

In (Actor-Critic) neither of the two smoothness constants are usually known in advance. The constant of the outer objective function depends on $\nabla_\pi Q^*(\pi)$, which is not known unless the solution of the inner problem is known exactly. In terms of the inner problem , one can show that the smoothness constant is the maximum eigenvalue of the matrix $(\gamma P^\pi - I)^2$, which requires additional computational burden to be estimated. On the contrary, we know the strong convexity constant of the inner problem, which is equal to $(1 - \gamma)$ because $P^\pi$ is a stochastic matrix and therefore its maximum eigenvalue equals to $1$.

We compare against two versions of (Ghadimi & Wang, 2018). One version, which is labelled as Ghadimi & Wang (tuned), tunes the step-sizes heuristically, while the other version, which is labelled as Ghadimi & Wang, uses the theoretical step-sizes calculated with an additional computational effort to estimate the hyper smoothness constant. As can be seen from Figure 2, our adaptive algorithm outperforms both variants of (Ghadimi & Wang, 2018). For completeness, we also compare against Natural Policy Gradient (Agarwal et al., 2020), labelled NPG and Policy Gradient (Sutton et al., 1999) labelled PG that are standard baselines in RL. Regarding the environment choice, we used *Gridworld* (Sutton & Barto, 2018) with 100 states.

We also compare the adaptive bilevel method proposed in our work with common benchmarks for bilevel problems. In particular, we tested with the two problems Covtype and the Hyper Data Cleaning problem on the Mnist dataset available in the Github repository https://github.com/benchopt/benchmark_bilevel. This repository also hosts numerous baselines that we can use as a comparison. In particular: SABA (Dagréou et al., 2022), AMIGO (Arbel & Mairal, 2021), BSA (Ghadimi & Wang, 2018), FSLA (Li et al., 2022), MRBO (Yang et al., 2021), VRBO (Yang et al., 2021) , TTSA (Hong et al., 2020) and SUSTAIN (Khanduri et al., 2021b). It turns out that on the Covtype experiment Figure 2c, our method is only slightly outperformed by AMIGO. However, AMIGO fails to converge in the Hyper Data Cleaning experiment ran on the Mnist dataset. In the Mnist experiment Figure 2d, all other methods compare similarly.

## 7 CONCLUSION

In this paper, we design a novel adaptive iterative method for possibly non-convex bilevel optimization problems. More precisely, we establish a two-level adaptive step-size schedule, which enables our method to simultaneously adjust its performance without any prior knowledge on the respective Lipschitz constants of both the inner and the outer problems. Moreover, we provide an asymptotic converge property of the method's iterates towards a solution. A natural question arises is whether we can design a similar accelerated adaptive algorithm for bilevel minimization with convex outer problem. We defer this question to some future work.

**Reproducibility Statement** The experimental details are provided in Appendix B and in the README of the attached code.

**Ethics Statement** The authors acknowledge that they have read and adhere to the ICLR Code of Ethics.

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

## A  ADDITIONAL RELATED WORK

A natural approach to overcome the above difficulty, which has been well-studied in the setting of "regular" optimization problems, is the so-called adaptive optimization method. In order to provide a quick overview, we start with the most popular adaptive method ADAGRAD of (Duchi et al., 2011) and (McMahan & Streeter, 2010), which is recently shown to interpolate between an $\mathcal{O}(1/\sqrt{T})$ and $\mathcal{O}(1/T)$ rate of convergence (Levy et al., 2018; Li & Orabona, 2019). More precisely, (Li & Orabona, 2019) shows that a specific, "one-lag-behind" variant of ADAGRAD with prior knowledge of the Lipschitz modulus achieves an $\mathcal{O}(1/T)$ rate in smooth, unconstrained problems; concurrently, (Levy et al., 2018) obtains the same rate in a parameter-agnostic context. The latter result is complemented by (Ward et al., 2019) also for non-convex objectives.

For the more relevant setting of smooth objective functions, adaptive methods can improve the rates from $\mathcal{O}(1/T)$ and achieve an accelerated $\mathcal{O}(1/T^2)$ rate, if convexity kicks in, by seamlessly adapting to the Lipschitz constant of the gradient, matching the well-known result in (Nesterov, 1983; 2004; Xiao, 2010). Three state-of-the-art methods of this type are ACCELEGRAD of (Levy et al., 2018), UNIXGRAD of (Kavis et al., 2019) and UNDERGRAD of (Antonakopoulos et al., 2022) achieve an $\mathcal{O}(1/\sqrt{T})$ rate of convergence for non-smooth or stochastic problems, and an $\mathcal{O}(1/T^2)$ rate of convergence for smooth optimization problems.

Furthermore, by employing an efficient line-search step, the *universal primal gradient descent* (UPGD) algorithm of (Nesterov, 2015) achieves order-optimal guarantees in the wider class of Hölder continuous problems (which includes the Lipschitz continuous and smooth cases as extreme cases). However, UPGD does not cover stochastic optimization problems or problems with relatively continuous / smooth objectives. Finally, (Antonakopoulos et al., 2020; Zhou et al., 2020a) are applicable in the same settings but are not adaptive, and they do not interpolate between different problem classes.

## B  EXPERIMENT DETAILS

**B.1. Details for RL experiment.** In the experiments we used a simple gridworld where the reward function equals $-1$ for any state action pairs and $0$ at the terminal state. The gridworld consists of 100 states arranged in a $10 \times 10$ grid with 4 actions corresponding to the four cardinal directions. For the the baselines, we use the following choices of step-sizes. For Ghadimi & Wang (tuned) we set the inner loop step-size $\eta = 0.9$ and the outer one as $\gamma = 0.5$. For Policy Gradient, we used step size equal to $0.75$ and for Natural Policy Gradient we used $3$. For both bilevel algorithms (Ours and (Ghadimi & Wang, 2018)), we obtained better results solving the inner loop more accurately choosing $t_k = 1000k$ and multiplying the theoretical step-size for the inner loop by a factor 10 and the outer step-size by 100. For what concerns compute resources, we run the experiments on a CPU equipped cluster. Moreover, one can notice that in the RL experiment, the hypergradient computation does not require solving a linear system. Indeed, the hyper-gradient entries equal to

$$\bar{\nabla} f(s,a) = -(1-\gamma)\bar{Q}^k(s,a)\rho^{\pi^k}(s)$$

as proven in (Hong et al., 2020, Section 4). In the above equation $\bar{Q}^k(s,a)$ is the approximate solution of the inner problem computed as illustrated in Algorithm 1 while $\rho^{\pi^k}$ is the occupancy measure induced by the policy $\pi^k$ which is the previous iterate of the outer loop. At this point, $\rho^{\pi^k}$ is the only unknown quantity and it can be computed with a dynamic programming procedures. In particular, notice that $\rho^{\pi^k}$ satisfies the following fixed point equation:

$$\rho^{\pi^k}(s) = (1-\gamma)\nu_0(s) + \gamma \sum_{\bar{s},\bar{a}} P(s|\bar{s},\bar{a})\pi^k(\bar{a}|\bar{s})\rho^{\pi^k}(\bar{s}).$$

Then, noticing that the operator $T : \Delta_S \to \Delta_S$, whose action is defined by

$$(T\rho)(s) = (1-\gamma)\nu_0(s) + \gamma \sum_{\bar{s},\bar{a}} P(s|\bar{s},\bar{a})\pi^k(\bar{a}|\bar{s})\rho(\bar{s}),$$

is a contraction. Hence, $||T^K u - \rho^{\pi^k}||_\infty \leq \gamma^K ||u - \rho^{\pi^k}||$.

Summarizing, in the experiments we approximated the occupancy measure in this way which enables us to avoid the expensive inversion of the linear system.

**B.2. Additional comparison with (Ghadimi & Wang, 2018).** The adaptive feature of our algorithm can be illustrated by providing two additional insights: in Figure 3 we report the same experiments but comparing only (Ghadimi & Wang, 2018) and our adaptive version. The advantage is dramatic in the Covtype experiment while in Mnist dataset the use of adaptive step-sizes does not improve the loss but also it does not worsen the performance of the well tuned algorithm by (Ghadimi & Wang, 2018).

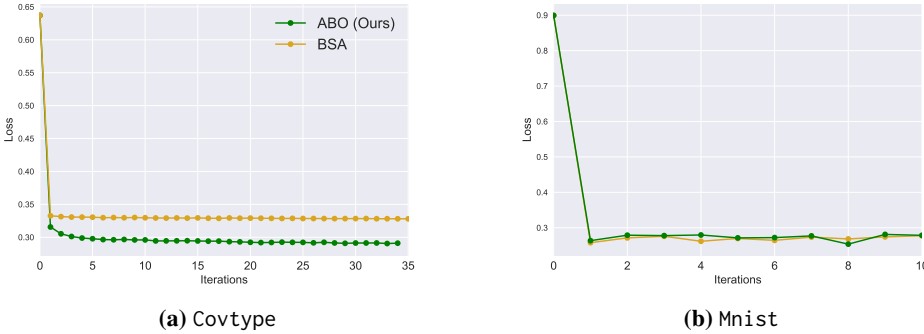

(a) Covtype                (b) Mnist

**Figure 3:** Experiments to show the advantage of the adaptive algorithm compared to its non adaptive counterpart (Ghadimi & Wang, 2018). We used the opensource benchmark available at https://github.com/benchopt/benchmark_bilevel on the dataset covtype and data cleaning experiment on mnist.

Finally, we carry out a study on the influence of the numerator of the outer step-size $\gamma_0$ on the performance on the Covtype benchmark and we notice that increasing it provides a better performance.

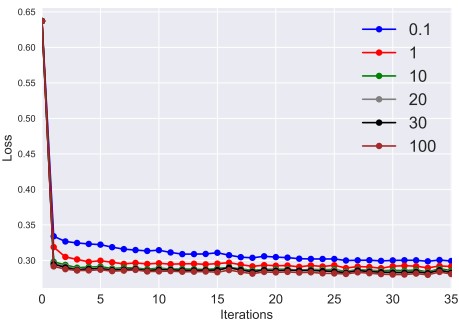

**Figure 4:** Study on the effect of $\gamma_0$

## C   DISTANCE GENERATING FUNCTIONS AND MIRROR MAPS

Our goal in this appendix is to derive some basic properties for the class of Bregman proximal mappings and mirror descent methods considered in the main body of our paper. Moreover, we draw connections with their respective "primal-dua" counterparts. Versions of the properties that we derive are known in the literature see, e.g., (Chen & Teboulle, 1993; Beck & Teboulle, 2003; Nesterov, 2009; Mertikopoulos & Sandholm, 2018; Héliou et al., 2020) and references therein. However, we include them for the sake of completeness.

To begin, we introduce two notions that will be particularly useful in the sequel. The first is the Fenchel conjugate of a Bregman function $h$, i.e.,

$$h^*(y) = \max_{x \in \mathrm{dom}\, h} \{\langle y, x \rangle - h(x)\}, \tag{C.1}$$

and the associated primal-dual *mirror map* $Q \colon \mathcal{V}^* \to \mathrm{dom}\, \partial h$:

$$Q(y) = \arg\max_{x \in \mathrm{dom}\, h} \{\langle y, x \rangle - h(x)\}. \tag{C.2}$$

That the above is well-defined is a consequence of the fact that $h$ is proper, l.s.c., convex and coercive;in addition, the fact that $Q$ takes values in $\mathrm{dom}\, \partial h$ follows from the fact that any solution of (C.2) must necessarily have nonempty subdifferential (see below). For completeness, we also recall here the definition of the Bregman proximal mapping

$$P_x(v) = \arg\min_{x' \in \mathrm{dom}\, h} \{\langle v, x - x' \rangle + D(x', x)\}, \tag{prox}$$

valid for all $x \in \mathrm{dom}\, \partial h$ and all $v \in \mathcal{V}^*$.

We then have the following basic lemma connecting the above notions:

**Lemma C.1.** *Let $h$ be a distance generating function with $K$-strong convexity modulus. Then, for all $x \in \mathrm{dom}\, \partial h$ and all $v, y \in \mathcal{V}^*$ we have:*

1. $x = Q(y) \iff y \in \partial h(x)$.

2. $x^+ = P_x(v) \iff \nabla h(x) + v \in \partial h(x) \iff x^+ = Q(\nabla h(x) + v)$.

3. *Finally, if $x = Q(y)$ and $p \in \mathcal{X}$, we get:*

$$\langle \nabla h(x), x - p \rangle \leq \langle y, x - p \rangle. \tag{C.3}$$

*Proof.* For the first equivalence, note that $x$ solves (C.1) if and only if $0 \in y - \partial h(x)$ and hence if and only if $y \in \partial h(x)$. Working in the same spirit for the second equivalence, we get that $x^+$ solves (prox) if and only if $\nabla h(x) + v \in \partial h(x^+)$ and therefore if and only if $x^+ = Q(\nabla h(x) + v)$.

For our last claim, by a simple continuity argument, it is sufficient to show that the inequality holds for the relative interior $\mathrm{ri}\, \mathcal{X}$ of $\mathcal{X}$ (which, in particular, is contained in $\mathrm{dom}\, \partial h$). In order to show this, pick a base point $p \in \mathrm{ri}\, \mathcal{X}$, and let

$$\phi(t) = h(x + t(p - x)) - [h(x) + \langle y, t(p - x) \rangle] \quad \text{for all } t \in [0, 1].$$

Since, $h$ is strongly convex and $y \in \partial h(x)$ due to the first equivalence, it follows that $\phi(t) \geq 0$ with equality if and only if $t = 0$. Since, $\psi(t) = \langle \nabla h(x + t(p - x)) - y, p - x \rangle$ is a continuous selection of subgradients of $\phi$ and both $\phi$ and $\psi$ are continuous over $[0, 1]$, it follows that $\phi$ is continuously differentiable with $\phi' = \psi$ on $[0, 1]$. Hence, with $\phi$ convex and $\phi(t) \geq 0 = \phi(0)$ for all $t \in [0, 1]$, we conclude that $\phi'(0) = \langle \nabla h(x) - y, p - x \rangle \geq 0$ and thus we obtain the result. $\qquad \square$

To proceed, the basic ingredient for establishing connections between Bregman proximal steps is a generalization of the rule of cosines which is known in the literature as the "three-point identity" (Chen & Teboulle, 1993). Using this tool, we will derive our main estimates.

**Lemma C.2.** *Let $h$ be a distance generating function. Then, for all $p \in \mathrm{dom}\, h$ and all $x, x' \in \mathrm{dom}\, \partial h$, we have*

$$D(p, x') = D(p, x) + D(x, x') + \langle \nabla h(x') - \nabla h(x), x - p \rangle.$$

Thanks to the three-point identity, we obtain the following estimate for the Bregman divergence before and after a mirror descent step:

**Proposition C.1.** *Let $h$ be a distance generating function with strong convexity modulus $K > 0$. Fix some $p \in \mathrm{dom}\, h$ and let $x^+ = P_x(v)$ for some $x \in \mathrm{dom}\, \partial h$ and $v \in \mathcal{V}^*$. We then have:*

$$D(p, x^+) \leq D(p, x) - D(x^+, x) + \langle v, x^+ - p \rangle \tag{C.4}$$

*and*

$$D(p, x^+) \leq D(p, x) + D(x, x^+) - \langle v, x - p \rangle. \tag{C.5}$$

*Proof.* By the three-point identity established in Lemma C.2, we have:

$$D(p, x) = D(p, x^+) + D(x^+, x) + \langle \nabla h(x) - \nabla h(x^+), x^+ - p \rangle$$

Rearranging terms then yields:

$$D(p, x^+) = D(p, x) - D(x^+, x) + \langle \nabla h(x^+) - \nabla h(x), x^+ - p \rangle.$$

By (C.3) and the fact that $x^+ = P_x(v)$ so $\nabla h(x) + v \in \partial h(x^+)$, the first inequality follows; the second one is obtained similarly. $\qquad\square$

Now, we change gears by introducing the necessary machinery for analysis primal-dual method. In particular, As we mentioned earlier, much of our analysis revolves around a "primal-dual" divergence between a target point $p \in \mathcal{X}$ and a dual vector $y \in \mathcal{Y}$, called the *Fenchel coupling*. Following (Mertikopoulos & Zhou, 2019), this is defined as follows for all $p \in \mathcal{X}, y \in \mathcal{Y}$:

$$F(p, y) = h(p) + h^*(y) - \langle y, p \rangle.$$

The following lemma illustrates some basic properties of the Fenchel coupling:

**Lemma C.3.** *Let $h$ be a distance generating function on $\mathcal{X}$ with convexity modulus $\alpha$. Then, for all $p \in \mathcal{X}$ and all $y \in \mathcal{Y}$, we have:*

1. *$F(p, y) = D(p, Q(y))$ if $Q(y) \in \mathcal{X}_\circ$ (but not necessarily otherwise).*

2. *If $x = Q(y)$, then $F(p, y) \geq \frac{K}{2} \|x - p\|^2$*

*Proof.* For our first claim, let $x = Q(y)$. Then, by definition we have:

$$F(p, y) = h(p) - \langle y, Q(y) \rangle - h(Q(y)) - \langle y, p \rangle = h(p) - h(x) - \langle y, p - x \rangle.$$

Since $y \in \partial h(x)$, we have $h'(x; p - x) = \langle y, p - x \rangle$ whenever $x \in \mathcal{X}_\circ$, thus proving our first claim. For our second claim, working in the previous spirit we get that:

$$F(p, y) = h(p) - h(x) - \langle y, p - x \rangle$$

Thus, we obtain the result by recalling the strong convexity assumption for $h$ with respect to the norm $\|\cdot\|$. $\qquad\square$

We continue with some basic relations connecting the Fenchel coupling relative to a target point before and after a gradient step. The basic ingredient for this is a primal-dual analogue of the so-called "three-point identity" for Bregman functions (Chen & Teboulle, 1993):

**Lemma C.4.** *Let $h$ be a distance generating function on $\mathcal{X}$. Fix some $p \in \mathcal{X}$ and let $y, y^+ \in \mathcal{Y}$. Then, letting $x = Q(y)$, we have*

$$F(p, y^+) = F(p, y) + F(x, y^+) + \langle y^+ - y, x - p \rangle. \tag{C.6}$$

*Proof.* By definition, we get:

$$F(p, y^+) = h(p) + h^*(y^+) - \langle y, p \rangle$$
$$F(p, y) = h(p) + h^*(y) - \langle y, p \rangle.$$

Then, by subtracting the above we get:

$$\begin{aligned} F(p, x^+) - F(p, y) &= h(p) + h^*(x^+) - \langle x^+, p \rangle - h(p) - h^*(y) + \langle y, p \rangle \\ &= h^*(x^+) - h^*(y) - \langle x^+ - y, p \rangle \\ &= h^*(x^+) - \langle y, Q(y) \rangle + h(Q(y)) - \langle x^+ - y, p \rangle \\ &= h^*(x^+) - \langle y, x \rangle + h(x) - \langle x^+ - y, p \rangle \\ &= h^*(x^+) + \langle x^+ - y, x \rangle - \langle x^+, x \rangle + h(x) - \langle x^+ - y, p \rangle \\ &= F(x, x^+) + \langle x^+ - y, x - p \rangle \end{aligned}$$

and our proof is complete.

$\qquad\square$

# D    Analysis of the inner loop

In this section, we focus on establishing the main convergence rates for the inner loop. We start by an intermediate lemma obtained in (Levy, 2017). In particular, we have the following:

**Lemma D.1.** *Let $f_t$ be a $\mu_t$- strongly convex function, for all $t \geq 0$. Moreover, assume that $\{x_t\}_{t \geq 0}$ are the iterates generated by*

$$x_{t+1} = x_t - \eta_t \nabla \tilde{f}_t(x_t) \tag{D.1}$$

*run with the adaptive step-size $\eta_t = \left( \sum_{s=1}^{t} \mu_s \right)^{-1}$. Then, the following regret bound holds*

$$\sum_{s=1}^{t} \left[ \tilde{f}_s(x_s) - \tilde{f}_s(x) \right] \leq \frac{1}{2} \sum_{s=1}^{t} \|\nabla \tilde{f}_s(x_s)\|_*^2. \tag{D.2}$$

Now, having the above generic regret bound, we are in position to provide our main convergence rate result regarding the algorithm's inner loop.

**Proposition D.1.** *Assume that $\{y_t\}_{t=0}^{t_k}$ are the iterates generated by Algorithm 1 inner loop and suppose that Assumption 2 holds true. Then, the following estimation holds*

$$\|\bar{y}_k - y^*(x_k)\| \leq \sqrt{\frac{2e(1 + t_k \mu_g / L_g)}{\mu_g^2 e^{t_k \mu_g / L_g}}}. \tag{D.3}$$

*Proof.* For the simplicity of the proof, we denote $a_{t_k} = \sum_{s=1}^{t_k} \|\nabla_y g(x_k; y_s)\|_*^{-2}$. Hence, we have

$$g(x_k; \bar{y}_k) - g(x_k; y^*(x_k)) \leq a_{t_k}^{-1} \sum_{s=1}^{t_k} \|\nabla_y g(x_k; y_s)\|_*^{-2} \left( g(x_k; y_s) - g(x_k; y^*(x_k)) \right).$$

From the $\mu_g$-strong convexity

$$g(x_k; y_s) - g(x_k; y^*(x_k)) \leq \langle \nabla_y g(x_k; y_s), y_s - y^*(x_k) \rangle - \frac{\mu_g}{2} \|y_s - y^*(x_k)\|^2.$$

Now, by setting

$$\tilde{f}_{s,k}(x) = \frac{1}{\|\nabla_y g(x_k; y_s)\|_*^2} \langle \nabla_y g(x_k; y_s), x \rangle + \frac{\mu_g}{2\|\nabla_y g(x_k; y_s)\|_*^2} \|x - y_s\|^2,$$

one may directly observe that the above sequence of functions are $\mu_g \|\nabla_y g(x_k; y_s)\|_*^{-2}$- strongly convex. Moreover, by combining the above relations we have

$$g(x_k; \bar{y}_k) - g(x_k; y^*(x_k)) \leq a_{t_k}^{-1} \sum_{s=1}^{t_k} \left[ \tilde{f}_{s,k}(y_s) - \tilde{f}_{s,k}(y^*(x_k)) \right]$$

$$\leq a_{t_k}^{-1} \sum_{s=1}^{t_k} \eta_s \|\nabla_y g(x_k, y_s)\|_*^{-2}$$

$$= a_{t_k}^{-1} \sum_{s=1}^{t_k} \frac{\|\nabla_y g(x_k; y_s)\|_*^{-2}}{\mu_g \sum_{j=1}^{s} \|\nabla_y g(x_k; y_s)\|_*^{-2}}$$

$$\leq \frac{a_{t_k}^{-1}}{\mu_g} \left( 1 + \log(\sum_{s=1}^{t_k} \|\nabla_y g(x_k; y_s)\|_*^{-2}) \right),$$

where the second inequality follows by combining Lemma D.1 and the fact that

$$\|\nabla \tilde{f}_{s,k}(y_s)\|_*^2 = \frac{1}{\|\nabla_y g(x_k; y_s)\|_*^4} \|\nabla_y g(x_k; y_s)\|_*^2 = \|\nabla_y g(x_k; y_s)\|_*^{-2}.$$

Finally, the last inequality is obtained from Lemma G.2. Therefore, by summarizing all the above, we have

$$g(x_k; \bar{y}_k) - g(x_k; y^*(x_k)) \leq \frac{1}{\mu_g} \left( \frac{1 + \log(\sum_{s=1}^{t_k} \|\nabla_y g(x_k; y_s)\|_*^{-2})}{\sum_{s=1}^{t_k} \|\nabla_y g(x_k; y_s)\|_*^{-2}} \right). \tag{D.4}$$

We shall now bound from above the (RHS) of the above inequality. First, we have

$$
\begin{aligned}
t_k &= \sum_{s=1}^{t_k} \frac{\|\nabla_y g(x_k; y_s)\|_*^2}{\|\nabla_y g(x_k; y_s)\|_*^2} \\
&\leq 2L_g \sum_{s=1}^{t_k} \frac{1}{\|\nabla_y g(x_k; y_s)\|_*^2} \left( g(x_k; y_s) - g(x_k; y^*(x_k)) \right) \\
&\leq 2L_g \sum_{s=1}^{t_k} \frac{1}{\|\nabla_y g(x_k; y_s)\|_*^2} \left( \langle \nabla_y g(x_k; y_s), y_s - y^*(x_k) \rangle - \frac{\mu_g}{2} \|y_s - y^*(x_k)\|^2 \right) \\
&\leq \frac{L_g}{\mu_g} \sum_{s=1}^{t_k} \frac{\|\nabla_y g(x_k; y_s)\|_*^{-2}}{\sum_{j=1}^{s} \|\nabla_y g(x_k; y_s)\|_*^{-2}} \\
&\leq \frac{L_g}{\mu_g} \left( 1 + \log(\sum_{s=1}^{t_k} \|\nabla_y g(x_k; y_s)\|_*^{-2}) \right),
\end{aligned}
$$

where the first inequality follows from the smoothness assumption, the second by the strong convexity, and the third inequality is obtained in the same manner as in the regret analysis above. Therefore, we have

$$
\frac{\mu_g}{L_g} t_k \leq 1 + \log(\sum_{s=1}^{t_k} \|\nabla_y g(x_k; y_s)\|_*^{-2}), \tag{D.5}
$$

which in turn yields

$$
1 \leq e^{t_k \mu_g / L_g} \leq e^{1 + \log(\sum_{s=1}^{t_k} \|\nabla_y g(x_k; y_s)\|_*^{-2})} = e \sum_{s=1}^{t_k} \|\nabla_y g(x_k; y_s)\|_*^{-2}.
$$

Moving forward, we observe that the function $z \to (1 + \log z)/z$ is decreasing for $z \geq 1$. Therefore

$$
\frac{1 + \log(e \sum_{s=1}^{t_k} \|\nabla_y g(x_k; y_s)\|_*^{-2})}{e \sum_{s=1}^{t_k} \|\nabla_y g(x_k; y_s)\|_*^{-2}} \leq \frac{1 + \log e^{t_k \mu_g / L_g}}{e^{t_k \mu_g / L_g}} = \frac{1 + t_k \mu_g / L_g}{e^{t_k \mu_g / L_g}}.
$$

Hence, we obtain

$$
\frac{1 + \log e + \log(\sum_{s=1}^{t_k} \|\nabla_y g(x_k; y_s)\|_*^{-2})}{\sum_{s=1}^{t_k} \|\nabla_y g(x_k; y_s)\|_*^{-2}} \leq e \frac{1 + t_k \mu_g / L_g}{e^{t_k \mu_g / L_g}}.
$$

So, a fortiori we have

$$
\frac{1 + \log(\sum_{s=1}^{t_k} \|\nabla_y g(x_k; y_s)\|_*^{-2})}{\sum_{s=1}^{t_k} \|\nabla_y g(x_k; y_s)\|_*^{-2}} \leq e \frac{1 + t_k \mu_g / L_g}{e^{t_k \mu_g / L_g}}.
$$

Thus, we have

$$
g(x_k; \bar{y}_k) - g(x_k; y^*(x_k)) \leq e \frac{1 + t_k \mu_g / L_g}{\mu_g e^{t_k \mu_g / L_g}}.
$$

Finally, due to the strong convexity, we have

$$
\begin{aligned}
g(x_k; \bar{y}_k) &\geq g(x_k; y^*(x_k)) + \langle \nabla_y g(x_k; y^*(x_k)), \bar{y}_k - y^*(x_k) \rangle + \frac{\mu_g}{2} \|\bar{y}_k - y^*(x_k)\|^2 \\
&= g(x_k; y^*(x_k)) + \frac{\mu_g}{2} \|\bar{y}_k - y^*(x_k)\|^2,
\end{aligned}
$$

where the last equality follows from the fact that $\nabla_y g(x_k; y^*(x_k)) = 0$ due to optimality of $y^*(x_k)$. Therefore, by summarizing we get

$$
\frac{\mu_g}{2} \|\bar{y}_k - y^*(x_k)\|^2 \leq e \frac{1 + t_k \mu_g / L_g}{\mu_g e^{t_k \mu_g / L_g}}.
$$

Then, the result follows by taking square roots on both sides. $\qquad \square$

# E    ANALYSIS OF THE OUTER LOOP

In this section, we shall provide the proofs for the convergence rate for the outer problem. We start with the summability of the adaptive residual. For the specific properties of the hypergradient, we refer the reader to Lemma 2.2 of (Ghadimi & Wang, 2018).

**Lemma E.1.** *Let $\{x_k\}_{k\geq 0}$ and $\{\bar{y}_k\}_{k\geq 0}$ be generated by Algorithm 1 with an inner loop length as in Theorem 1. Then, the following hold:*

1. *The residual $\delta_k^2$ is summable, i.e.,*
$$\sum_{k=1}^{+\infty} \delta_k^2 < +\infty.$$

2. *The step-size $\gamma_k$ converges to its strictly positive infimum, i.e.,*
$$\lim_k \gamma_k = \gamma_\infty > 0.$$

*Proof.* From the smoothness of $f$, we have

$$
\begin{aligned}
f(x_{k+1}; y^*(x_{k+1})) &\leq f(x_k; y^*(x_k)) + \langle \nabla f(x_k; y^*(x_k)), x_{k+1} - x_k \rangle + \frac{L_f}{2}\|x_{k+1} - x_k\|^2 \\
&= f(x_k; y^*(x_k)) + \langle \nabla f(x_k; y^*(x_k)) - \tilde{\nabla} f(x_k; \bar{y}_k), x_{k+1} - x_k \rangle \\
&\quad + \langle \tilde{\nabla} f(x_k; \bar{y}_k), x_{k+1} - x_k \rangle + \frac{L_f}{2}\|x_{k+1} - x_k\|^2 \\
&\leq f(x_k; y^*(x_k)) + \|\nabla f(x_k; y^*(x_k)) - \tilde{\nabla} f(x_k; \bar{y}_k)\| \cdot \|x_{k+1} - x_k\| \\
&\quad + \langle \tilde{\nabla} f(x_k; \bar{y}_k), x_{k+1} - x_k \rangle + \frac{L_f}{2}\|x_{k+1} - x_k\|^2,
\end{aligned}
$$

where the last inequality follows from the Cauchy-Schwartz inequality. Moreover, by using the properties of the hypergradient we have

$$
\begin{aligned}
\|\nabla f(x_k; y^*(x_k)) - \tilde{\nabla} f(x_k; \bar{y}_k)\| \cdot \|x_{k+1} - x_k\| &\leq L_f \|y^*(x_k) - \bar{y}_k\| \cdot \|x_{k+1} - x_k\| \\
&\leq \frac{L_f}{2}\|y^*(x_k) - \bar{y}_k\|^2 + \frac{L_f}{2}\|x_{k+1} - x_k\|^2,
\end{aligned}
$$

where the last inequality follows from the Fenchel-Young inequality. Combining together yields

$$
\begin{aligned}
f(x_{k+1}; y^*(x_{k+1})) &\leq f(x_k; y^*(x_k)) + \frac{L_f}{2}\|y^*(x_k) - \bar{y}_k\|^2 + L_f\|x_{k+1} - x_k\|^2 \\
&\quad + \langle \tilde{\nabla} f(x_k; \bar{y}_k), x_{k+1} - x_k \rangle \\
&\leq f(x_k; y^*(x_k)) + \frac{L_f}{2}\|y^*(x_k) - \bar{y}_k\|^2 + \frac{2L_f}{K_h} F(x_{k+1}, U_k) \\
&\quad + \langle \tilde{\nabla} f(x_k; \bar{y}_k), x_{k+1} - x_k \rangle,
\end{aligned}
$$

where the last inequality follows from Lemma C.3. Focusing on the last term of the (RHS) we have:

$$
\begin{aligned}
\langle \tilde{\nabla} f(x_k; \bar{y}_k), x_{k+1} - x_k \rangle &= \frac{1}{\gamma_k} \langle U_k - U_{k+1}, x_{k+1} - x_k \rangle \\
&= -\frac{1}{\gamma_k}\left( F(x_k, U_{k+1}) + F(x_{k+1}, U_k) \right) \\
&= -\frac{1}{\gamma_k}\gamma_k^2 \delta_k^2 \\
&= -\gamma_k \delta_k^2.
\end{aligned}
$$

Hence,

$$
f(x_{k+1}; y^*(x_{k+1})) \leq f(x_k; y^*(x_k)) + \frac{L_f}{2}\|y^*(x_k) - \bar{y}_k\|^2 + \frac{2L_f}{K_h} F(x_{k+1}, U_k) - \gamma_k \delta_k^2.
$$

Moreover, by the definition of $\delta_k$ and the fact that the Fenchel coupling is non-negative, we have

$$F(x_{k+1}, U_k) \leq F(x_{k+1}, U_k) + F(x_k, U_{k+1}) = \delta_k^2 \gamma_k^2,$$

which combined with the above inequality implies that

$$f(x_{k+1}; y^*(x_{k+1})) \leq f(x_k; y^*(x_k)) + \frac{L_f}{2} \cdot \frac{2e(1 + t_k \mu_g/L_g)}{\mu_g^2 e^{t_k \mu_g/L_g}} + \frac{2L_f}{K_h} \delta_k^2 \gamma_k^2 - \gamma_k \delta_k^2,$$

where we also used Proposition D.1. Thus

$$\frac{1}{2}\gamma_k \delta_k^2 \leq f(x_k; y^*(x_k)) - f(x_{k+1}; y^*(x_{k+1})) + \frac{L_f e(1 + t_k \mu_g/L_g)}{\mu_g^2 e^{t_k \mu_g/L_g}} + \left( \frac{2L_f}{K_h} \gamma_k - \frac{1}{2} \right) \gamma_k \delta_k^2. \quad \text{(E.1)}$$

Now, since $\gamma_k \geq 0$ is a decreasing sequence, then its limit exists and more precisely we have

$$\lim_k \gamma_k = \gamma_\infty \geq 0.$$

We assume that $\gamma_\infty = 0$. Then, combining the fact that $\gamma_k$ is vanishing and (E.1), we get that there exists some $k_0 \in \mathbb{N}$ such that

$$\gamma_k \leq \frac{K}{2L_f} \quad \text{for all } k \geq k_0,$$

which in turns yields that for all $k \geq k_0$ we have

$$\frac{1}{2}\gamma_k \delta_k^2 \leq f(x_k; y^*(x_k)) - f(x_{k+1}; y^*(x_{k+1})) + \frac{L_f e(1 + t_k \mu_g/L_g)}{\mu_g^2 e^{t_k \mu_g/L_g}}.$$

In what follows, without the loss of generality, we shall assume that $k_0 = 1$. Therefore, after telescoping, we have

$$\frac{1}{2}\sum_{k=1}^T \gamma_k \delta_k^2 \leq f(x_1; y^*(x_1)) - \min_{x \in \mathcal{X}} f(x; y^*(x)) + \sum_{k=1}^T \frac{L_f e(1 + t_k \mu_g/L_g)}{\mu_g^2 e^{t_k \mu_g/L_g}}.$$

Now, by bounding from below the (LHS) we have

$$\sum_{k=1}^T \gamma_k \delta_k^2 = \sum_{k=1}^T \frac{\delta_k^2}{\sqrt{1 + \sum_{j=1}^{k-1} \delta_j^2}}$$

$$\geq \left( \sum_{k=1}^T \delta_k^2 \right) \frac{1}{\sqrt{1 + \sum_{k=1}^T \delta_k^2}}$$

$$= \left( 1 + \sum_{k=1}^T \delta_k^2 - 1 \right) \frac{1}{\sqrt{1 + \sum_{k=1}^T \delta_k^2}}$$

$$= \sqrt{1 + \sum_{k=1}^T \delta_k^2} - \frac{1}{\sqrt{1 + \sum_{k=1}^T \delta_k^2}}.$$

Hence, after rearranging we have

$$\sqrt{1 + \sum_{k=1}^T \delta_k^2} \leq f(x_1; y^*(x_1)) - \min_{x \in \mathcal{X}} f(x; y^*(x)) + \frac{L_f e(1 + t_k \mu_g/L_g)}{\mu_g^2 e^{t_k \mu_g/L_g}} + \frac{1}{\sqrt{1 + \sum_{k=1}^T \delta_k^2}}$$

$$\leq f(x_1; y^*(x_1)) - \min_{x \in \mathcal{X}} f(x; y^*(x)) + \frac{L_f e(1 + t_k \mu_g/L_g)}{\mu_g^2 e^{t_k \mu_g/L_g}} + 1.$$

So, we finally get that

$$\frac{1}{\gamma_{T+1}} \leq f(x_1; y^*(x_1)) - \min_{x \in \mathcal{X}} f(x; y^*(x)) + \frac{L_f e(1 + t_k \mu_g/L_g)}{\mu_g^2 e^{t_k \mu_g/L_g}} + 1.$$

Therefore, by letting $T \to +\infty$ and recalling the fact that $\lim_T \gamma_T = 0$, we get that

$$+\infty = \lim_T \frac{1}{\gamma_{T+1}} \leq f(x_1; y^*(x_1)) - \min_{x \in \mathcal{X}} f(x; y^*(x)) + 1.$$

So, we get $+\infty < +\infty$ which is a contradiction. So, we get $\lim_k \gamma_k = \gamma_\infty > 0$. This in turn yields

$$\sum_{k=1}^{+\infty} \delta_k^2 = \lim_{T \to +\infty} \sum_{k=1}^{T} \delta_k^2 = \lim_{T \to +\infty} \left( \frac{1}{\gamma_T^2 - 1} \right) = \frac{1}{\gamma_\infty} - 1 < +\infty,$$

since $\gamma_\infty > 0$, and hence the result follows. $\qquad\square$

We move forward now by showing that $\sup_{x^* \in \mathcal{X}^*} F(x^*, x_k)$ is uniformly bounded.

**Lemma E.2.** *Let $\{x_k\}_{k \geq 0}$ and $\{\bar{y}_k\}_{k \geq 0}$ be generated by* Algorithm 1*. Then, the sequence,*

$$\sup_{x^* \in \mathcal{X}^*} F(x^*, U_k), \quad k \geq 0,$$

*is bounded.*

*Proof.* Working in the previous spirit, we have that:

$$F(x^*, U_{k+1}) \leq F(x^*, U_k) + \gamma_k \delta_k^2 + \|\nabla f(x_k; y^*(x_k)) - \tilde{\nabla} f(x_k; \bar{y}_k)\| \|x_k - x^*\|.$$

for all $x^* \in \mathcal{X}^*$. The above in turn yields:

$$F(x^*, U_{k+1}) \leq F(x^*, U_k) + \gamma_k^2 \delta_k^2 + L_f \operatorname{diam} \mathcal{X} \sqrt{\frac{2e(1 + t_k \mu_g/L_g)}{\mu_g^2 e^{t_k \mu_g/L_g}}}.$$

Now, after telescoping we get:

$$F(x^*, U_{T+1}) \leq F(x^*, U_1) + \sum_{k=1}^{T} \gamma_k^2 \delta_k^2 + \sum_{k=1}^{T} L_f \operatorname{diam} \mathcal{X} \sqrt{\frac{2e(1 + t_k \mu_g/L_g)}{\mu_g^2 e^{t_k \mu_g/L_g}}},$$

and hence by applying Lemma E.1, we readily get that

$$F(x^*, U_{T+1}) \leq F(x^*, U_1) + \sum_{k=1}^{+\infty} \gamma_k^2 \delta_k^2 + \sum_{k=1}^{+\infty} L_f \operatorname{diam} \mathcal{X} \sqrt{\frac{2e(1 + t_k \mu_g/L_g)}{\mu_g^2 e^{t_k \mu_g/L_g}}} < +\infty.$$

The result follows by taking suprema on both sides relative to the solution set $\mathcal{X}^*$. $\qquad\square$

Finally, having the above results we are able to provide proof of our main convergence rate result.

**Theorem E.1.** *Let $\{x_k\}_{k \geq 0}$ and $\{\bar{y}_k\}_{k \geq 0}$ be generated by* Algorithm 1 *Then, the following convergence rates hold:*

1. *If $f$ is convex, then*
$$f(\bar{x}_T; y^*(\bar{x}_T)) - \min_{x \in \mathcal{X}} f(x; y^*(x)) = \mathcal{O}(1/T).$$

2. *If $f$ is non-convex, then*
$$\min_{1 \leq k \leq T} \|\nabla f(x_k; y^*(x_k))\|^2 = \mathcal{O}(1/T).$$

*Proof.* For the convex case, we have:

$$\begin{aligned}
f(x_k; y^*(x_k)) - f(x^*; y^*(x^*)) &\leq \langle \nabla f(x_k; y^*(x_k)), x_k - x^* \rangle \\
&= \langle \nabla f(x_k; y^*(x_k)) - \tilde{\nabla} f(x_k; \bar{y}_k), x_k - x^* \rangle \\
&\quad + \langle \tilde{\nabla} f(x_k; \bar{y}_k), x_k - x^* \rangle.
\end{aligned}$$

Therefore, following the same approach as above, we have

$$f(x_k; y^*(x_k)) - f(x^*; y^*(x^*)) \leq \frac{1}{\gamma_k} \langle U_k - U_{k+1}, x_k - x^* \rangle$$
$$+ \|\nabla f(x_k; y^*(x_k)) - \tilde{\nabla} f(x_k; \bar{y}_k)\| \|x_k - x^*\|,$$

which by employing Lemma C.4 yields

$$f(x_k; y^*(x_k)) - f(x^*; y^*(x^*)) \leq \frac{1}{\gamma_k} \left( F(x^*, U_k) - F(x^*, U_{k+1}) + \gamma_k \delta_k^2 \right)$$
$$+ \|\nabla f(x_k; y^*(x_k)) - \tilde{\nabla} f(x_k; \bar{y}_k)\| \|x_k - x^*\|.$$

So, by applying Lemma C.4 we have

$$f(x_k; y^*(x_k)) - f(x^*; y^*(x^*)) \leq \frac{1}{\gamma_k} \left( F(x^*, U_k) - F(x^*, U_{k+1}) + F(x_k, U_k) + F(x_{k+1}, U_k) \right)$$
$$L_f \|y^*(x_k) - \bar{y}_k\| \operatorname{diam} \mathcal{X},$$

and so,

$$f(x_k; y^*(x_k)) - f(x^*; y^*(x^*)) \leq \frac{F(x^*, U_k)}{\gamma_k} - \frac{F(x^*, U_{k+1})}{\gamma_{k+1}}$$
$$\left( \frac{1}{\gamma_{k+1}} - \frac{1}{\gamma_k} \right) F(x^*, U_{k+1}) + \gamma_k \delta_k^2 + L_f \operatorname{diam} \mathcal{X} \sqrt{\frac{2e(1 + t_k \mu_g/L_g)}{\mu_g^2 e^{t_k \mu_g/L_g}}}.$$

Now, by telescoping and applying the fact that $\sup_{x^* \in \mathcal{X}^*} F(x^*, U_k)$ is (uniformly) bounded, i.e., $\sup_{x^* \in \mathcal{X}^*} F(x^*, U_k) \leq D$, we get

$$f(\bar{x}_T; y^*(\bar{x}_T)) - f(x^*; y^*(x^*)) \leq \frac{1}{T} \left( \frac{D}{\gamma_{T+1}} + \sum_{k=1}^{T} \gamma_k \delta_k^2 \right.$$
$$\left. + L_f \operatorname{diam} \mathcal{X} \sum_{k=1}^{T} \sqrt{\frac{2e(1 + \mu_g/L_g t_k)}{\mu_g^2 e^{\mu_g/L_g t_k}}} \right),$$

and by recalling the previous analysis the result follows since the (RHS) of the above is bounded. Now, we turn our attention to the non-convex case. By invoking Lemma E.1 for the unconstrained case, we have

$$f(x_{k+1}; y^*(x_{k+1})) \leq f(x_k; y^*(x_k)) + \langle \nabla f(x_k; y^*(x_k)), x_{k+1} - x_k \rangle + \frac{L_f}{2} \|x_{k+1} - x_k\|^2,$$

and moreover, we have

$$f(x_{k+1}; y^*(x_{k+1})) \leq f(x_k; y^*(x_k)) - \langle \nabla f(x_k; y^*(x_k)), \tilde{\nabla} f(x_k; \bar{y}_k) \rangle + \frac{L_f}{2} \|x_{k+1} - x_k\|^2,$$

which in turn yields

$$f(x_{k+1}; y^*(x_{k+1})) \leq f(x_k; y^*(x_k)) - \gamma_k \langle \nabla f(x_k; y^*(x_k)), \tilde{\nabla} f(x_k; \bar{y}_k) - \nabla f(x_k; y^*(x_k)) \rangle$$
$$- \gamma_k \|\nabla f(x_k; y^*(x_k))\|^2 + \frac{L_f}{2} \|x_{k+1} - x_k\|^2.$$

Therefore, following the same steps as above, after rearranging and telescoping we get

$$\sum_{k=1}^{T} \gamma_k \|\nabla f(x_k; y^*(x_k))\|^2 \leq f(x_1; y^*(x_1)) - \min_{x \in \mathcal{X}} f(x; y^*(x)) + \frac{L_f}{2} \sum_{k=1}^{T} \|y^*(x_k) - \bar{y}_k\|^2$$
$$+ \frac{L_f}{2} \sum_{k=1}^{T} \|x_{k+1} - x_k\|^2.$$

Now, by combining the fact that $\gamma_k \to \inf_k \gamma_k = \gamma_\infty > 0$ and the summability of the respective residual we obtain

$$\gamma_\infty \sum_{k=1}^{T} \|\nabla f(x_k; y^*(x_k))\|^2 \leq f(x_1; y^*(x_1)) - \min_{x \in \mathcal{X}} f(x; y^*(x)) + \frac{L_f}{2} \sum_{k=1}^{+\infty} \|y^*(x_k) - \bar{y}_k\|^2$$
$$+ \frac{L_f}{2} \sum_{k=1}^{+\infty} \|x_{k+1} - x_k\|^2.$$

Hence, we in turn get

$$\min_{1 \leq k \leq T} \|\nabla f(x_k; y^*(x_k))\|^2 \leq \frac{1}{\gamma_\infty T} \left( f(x_1; y^*(x_1)) - \min_{x \in \mathcal{X}} f(x; y^*(x)) + \frac{L_f}{2} \sum_{k=1}^{+\infty} \|y^*(x_k) - \bar{y}_k\|^2 \right.$$
$$\left. + \frac{L_f}{2} \sum_{k=1}^{+\infty} \|x_{k+1} - x_k\|^2. \right.$$

and hence the result follows. □

# F    LAST ITERATE CONVERGENCE RESULT

In this section, we show the asymptotic convergence of the actual iterates of Algorithm 1, dubbed as the last iterate convergence. To that end, for this section, we will make the following weak-secant inequality. Throughout this section, we shall assume that the respective distance generating function $h$ satisfies the so-called *reciprocity condition*. In particular, note that if $F(x^*, U_k) \to 0$ then we readily have that $x_k = Q(U_k) \to x^*$. This is obtained by (C.3) which yields that:

$$F(x^*, U_k) \leq \frac{K}{2} \|Q(U_k) - x^*\|^2.$$

The reciprocity condition dictates that also the converse is also true. More precisely, we make the following assumption:

$$Q(U_k) \to x^*, \quad \text{then} \quad F(x^*, U_k) \to 0.$$

First, we show that the sequence $\{x_k\}_{k \geq 0}$ has a convergent subsequence to a point the solution set $\mathcal{X}^*$. In doing so, we first show that the said sequence admits at least one limit point. Formally, this is given by the following lemma.

**Lemma F.1.** *Assume that $\{x_k\}_{k \geq 0}$ is generated by Algorithm 1 with an inner loop length $t_k$ as in Theorem 1. Then, it admits at least one limit point.*

*Proof.* Following the same path, we have that

$$F(x^*, U_{k+1}) \leq F(x^*, U_k) + \frac{2e\left(1 + t_k \mu_g / L_g\right)}{\mu_g e^{t_k \mu_g / L_g}} + \gamma_k^2 \delta_k^2.$$

Therefore, by telescoping we have

$$F(x^*, U_{k+1}) \leq F(x^*, U_0) + \sum_{k=1}^{T} \frac{2e\left(1 + t_k \mu_g / L_g\right)}{\mu_g e^{t_k \mu_g / L_g}} + \sum_{k=1}^{T} \gamma_k \delta_k^2,$$

which in turn yields that the sequence $F(x^*, U_{k+1})$ is bounded by invoking the summability obtained in Lemma E.1. So, we have that for all $x^* \in \mathcal{X}^*$

$$\|x_k\|^2 \leq 2\|x_k - x^*\|^2 + 2\|x^*\|^2 \leq \frac{4}{K} F(x^*, U_k) + 2\|x^*\|^2,$$

which in turns yields that $\{x_k\}_{k \geq 0}$ is bounded and therefore it admits at least one limit point. □

**Lemma F.2.** *Assume that $\{x_k\}_{k\geq 0}$ is generated by Algorithm 1 with an inner loop length $t_k$ as in Theorem 1. Then, there exists a subsequence $\{x_{k_n}\}_{n\geq 0}$, which converges to $\mathcal{X}^*$.*

*Proof.* Assume to the contrary that such a sub-sequence does not exist, ie $x_k$ admits no limit points which belong to the solution set $\mathcal{X}^*$. Then, there exists a non-empty closed set $\mathcal{K} \subset \mathcal{X}$ which is separated by neighbourhoods from $\mathcal{X}^*$ such that $x_k \in \mathcal{K}$ for $k$ large enough. Then, by relabelling $x_k$, if necessary, we may assume that $x_k \in \mathcal{K}$ for all $k$.

Thus, we have for all $x^* \in \mathcal{X}^*$, that

$$F(x^*, U_{k+1}) \leq F(x^*, U_k) - \gamma_k \langle \tilde{\nabla} f(x_k; y^*(x_k)), x_k - x^* \rangle + \gamma_k^2 \delta_k^2,$$

which in turn yields

$$F(x^*, U_{k+1}) \leq F(x^*, U_k) - \gamma_k \langle \tilde{\nabla} f(x_k; \bar{y}_k) - \nabla f(x_k; y^*(x_k)), x_k - x^* \rangle + \gamma_k^2 \delta_k^2$$
$$- \gamma_k \langle \nabla f(x_k; y^*(x_k)), x_k - x^* \rangle + \gamma_k^2 \delta_k^2.$$

So, we have

$$F(x^*, U_{k+1}) \leq F(x^*, U_k) + \gamma_k \|\tilde{\nabla} f(x_k; \bar{y}_k) - \nabla f(x_k; y^*(x_k))\| \operatorname{diam} \mathcal{X}$$
$$- \gamma_k \inf\{\langle \nabla f(x; y^*(x)), x - x^* \rangle : x^* \in \mathcal{X}^*, x \in \mathcal{K}\} + \gamma_k^2 \delta_k^2.$$

Now, by applying weak secant inequality we get

$$F(x^*, U_{k+1}) \leq F(x^*, U_k) + L_f \operatorname{diam} \mathcal{X} \|y^*(x_k) - \bar{y}_k\| - \gamma_k \varepsilon(\mathcal{K}) + \gamma_k^2 \delta_k^2,$$

where:

$$\inf\{\langle \nabla f(x; y^*(x)), x - x^* \rangle : x^* \in \mathcal{X}^*, x \in \mathcal{K}\} = \varepsilon(\mathcal{K}) > 0$$

Now, after telescoping and factoring out $\sum_{k=1}^{T} \gamma_k$ we have

$$F(x^*, U_{T+1}) \leq F(x^*, U_0) + \sum_{k=1}^{T} \gamma_k \left( L_f \operatorname{diam} \mathcal{X} \sum_{k=1}^{T} \|y^*(x_k - \bar{y}_k)\| \left(\sum_{k=1}^{T} \gamma_k\right)^{-1} - \varepsilon(\mathcal{K}) \right.$$
$$\left. + \sum_{k=1}^{T} \gamma_k^2 \delta_k^2 \left(\sum_{k=1}^{T} \gamma_k\right)^{-1} \right).$$

Now, dealing with the terms on the (RHS) and computing their limits while $T \to +\infty$ we get

- For the term $\sum_{k=1}^{T} \gamma_k$ we have

$$\sum_{k=1}^{T} \gamma_k \geq \gamma_\infty T \to +\infty$$

  since $\gamma_\infty > 0$, by invoking Lemma E.1.

- For the term $L_f \operatorname{diam} \mathcal{X} \sum_{k=1}^{T} \|y^*(x_k - \bar{y}_k)\| \left(\sum_{k=1}^{T} \gamma_k\right)^{-1}$ we have that:

$$L_f \operatorname{diam} \mathcal{X} \sum_{k=1}^{T} \|y^*(x_k - \bar{y}_k)\| \left(\sum_{k=1}^{T} \gamma_k\right)^{-1} \leq \frac{L_f \operatorname{diam} \mathcal{X} \sum_{k=1}^{+\infty} \|y^*(x_k - \bar{y}_k)\|}{\gamma_\infty T}$$

  and since the nominator is bounded we readily get that $L_f \operatorname{diam} \mathcal{X} \sum_{k=1}^{T} \|y^*(x_k - \bar{y}_k)\| \left(\sum_{k=1}^{T} \gamma_k\right)^{-1} \to 0$ while $T \to +\infty$.

- Finally for the term $\sum_{k=1}^{T} \gamma_k^2 \delta_k^2 \left(\sum_{k=1}^{T} \gamma_k\right)^{-1}$ we have:

$$\sum_{k=1}^{T} \gamma_k^2 \delta_k^2 \left(\sum_{k=1}^{T} \gamma_k\right)^{-1} \leq \frac{\sum_{k=1}^{+\infty} \gamma_k^2 \delta_k^2}{\gamma_\infty T}$$

  which again yields if one invokes Lemma E.1 that $\sum_{k=1}^{T} \gamma_k^2 \delta_k^2 \left(\sum_{k=1}^{T} \gamma_k\right)^{-1} \to 0$ while $T \to +\infty$.

Therefore summarizing we get that:

$$\lim_T F(x^*, U_{T+1}) \leq F(x^*, U_0) - \infty \varepsilon(\mathcal{K}) = -\infty,$$

which is a contradiction since $F(x^*, U_{T+1}) \geq 0$, and hence the result follows. $\qquad \square$

Having all this at hand, we are finally in the position to prove the convergence of the actual iterates of the method. In order to accomplish this, we need an intermediate lemma that allows us to move from a convergent subsequence to a global convergence (see also Combettes (2001), Polyak (1987)).

**Lemma F.3.** *Let* $\chi \in (0, 1]$, $\{\alpha_k\}_{k \geq 0}$ *and* $\{\beta_k\}_{k \geq 0}$ *are sequences of non-negative numbers, and* $\{\varepsilon_k\}_{k \geq 0} \in l^1(\mathbb{N})$. *Suppose that for all* $k = 1, 2, \ldots,$ *we have*

$$\alpha_{k+1} \leq \chi \alpha_k - \beta_t + \varepsilon_k. \tag{F.1}$$

*Then,* $\{\alpha_k\}_{k \geq 0}$ *converges.*

Moreover, the following lemma establishes the convergence to a point result for quasi-Fejer sequences, i.e., sequences satisfying (F.1).

**Lemma F.4.** *Assume* $\{x_k\}_{k \geq 0}$ *is a quasi-Fejer sequence relative to the solution set* $\mathcal{X}^*$, *i.e., for all* $x^* \in \mathcal{X}^*$

$$\|x_{k+1} - x^*\|^2 \leq \|x_k - x^*\|^2 + \varepsilon_k,$$

*with* $\{\varepsilon_k\}_{k \geq 0}$ *being a non-negative summable sequence.*

*Moreover, assume that every limit point of* $\{x_k\}_{k \geq 0}$ *belongs to* $\mathcal{X}^*$. *Then,* $\{x_k\}_{k \geq 0}$ *converges to some* $x^* \in \mathcal{X}^*$.

Now that we have established the above intermediate result, we can prove the main result.

**Theorem F.1.** *Assume that* $\{x_k\}_{k \geq 0}$ *is generated by Algorithm 1 with an inner loop length* $t_k$ *as in Theorem 1. Then, the following properties hold*

1. *The sequence* $\{\mathrm{dist}(x_k, \mathcal{X}^*)\}_{k \geq 0}$ *converges to* $0$.

2. *If* $\mathcal{X}$ *is compact, then* $\{x_k\}_{k \geq 0}$ *converges to a point* $x^* \in \mathcal{X}^*$

*Proof.* We start with the first claim. In particular, by applying Lemma F.3, we straightforwardly obtain that $\{\inf_{x^* \in \mathcal{X}^*} F(x^*, x_k)\}_{k \geq 0}$ is convergent. Thus, by applying the reciprocity condition (F), we immediately get that $\inf_{x^* \in \mathcal{X}^*} \|x_k - x^*\|^2$ is also convergent.

Thus, since there exists a convergent subsequence of $\{x_k\}_{k \geq 0}$ to the solution set $\mathcal{X}^*$, then the first claim follows.

Moving forward to the second claim, we observe that due to compactness, the set of limit points of the sequence $\{x_k\}_{k \geq 0}$ is non-empty. Let $x_\infty$ be a limit point of $\{x_k\}_{k \geq 0}$. Then, there exists a subsequence $x_{\lambda_k}$ such that $x_{\lambda_k} \to x_\infty$ as $k \to \infty$. Moreover, by invoking continuity arguments we have that

$$\lim_{k \to \infty} \inf_{x^* \in \mathcal{X}^*} \|x_k - x^*\|^2 = \inf_{x^* \in \mathcal{X}^*} \|x_\infty - x^*\|^2 = 0,$$

which yields that $x_\infty \in \mathcal{X}^*$ by the closedness of $\mathcal{X}^*$. Thus, since the above analysis holds for all limit points of $\{x_k\}_{k \geq 0}$, we get that every limit point belongs to $\mathcal{X}^*$. Therefore, since $\{x_k\}_{k \geq 0}$ is a quasi-Fejer sequence, the second claim follows. $\qquad \square$

## G  LEMMAS ON NUMERICAL SEQUENCES

In this appendix, we provide some necessary inequalities on numerical sequences that we require for the convergence rate analysis of the previous sections. Most of the lemmas presented below already exist in the literature, and go as far back as Auer et al. (2002) and McMahan & Streeter (2010); when appropriate, we note next to each lemma the references with the statement closest to the precise version we are using in our analysis. These lemmas can also be proved by the general methodology outlined in Gaillard et al. (2014, Lemma 14), so we only provide a proof for two ancillary results that would otherwise require some more menial bookkeeping.

**Lemma G.1** ([McMahan & Streeter](), 2010, [Levy et al.](), 2018). *For all non-negative numbers $\alpha_1, \ldots \alpha_t$, the following inequality holds*

$$\sqrt{\sum_{t=1}^{T} \alpha_t} \leq \sum_{t=1}^{T} \frac{\alpha_t}{\sqrt{\sum_{i=1}^{t} \alpha_i}} \leq 2\sqrt{\sum_{t=1}^{T} \alpha_t}.$$

**Lemma G.2** ([Levy et al.](), 2018). *For all non-negative numbers $\alpha_1, \ldots \alpha_t$, the following inequality holds:*

$$\sum_{t=1}^{T} \frac{\alpha_t}{1 + \sum_{i=1}^{t} \alpha_i} \leq 1 + \log(1 + \sum_{t=1}^{T} \alpha_t).$$

**Lemma G.3.** *Let $b_1, \ldots, b_t$ non-negative numbers with $b_1 > 0$. Then, the following inequality holds:*

$$\sum_{t=1}^{T} \frac{b_t}{\sum_{i=1}^{t} b_i} \leq 2 + \log\left(\frac{\sum_{t=1}^{T} b_t}{b_1}\right).$$

*Proof.* It is immediately follows from [Lemma G.2]() with $\alpha_t = b_t/b_1$. $\qquad\square$

The following set of inequalities is due to [Bach & Levy]() (2019). For completeness, we provide a sketch of their proof.

**Lemma G.4** ([Bach & Levy](), 2019). *For all non-negative numbers $\alpha_1, \ldots \alpha_t \in [0, \alpha]$ with $\alpha_0 > 0$, the following inequalities hold:*

$$\sqrt{\alpha_0 + \sum_{t=1}^{T-1} \alpha_i} - \sqrt{\alpha_0} \leq \sum_{t=1}^{T} \frac{\alpha_t}{\sqrt{\alpha_0 + \sum_{i=1}^{t-1} \alpha_j}} \leq \frac{2\alpha}{\sqrt{\alpha_0}} + 3\sqrt{\alpha} + 3\sqrt{\alpha_0 + \sum_{t=1}^{T-1} \alpha_t}.$$

**Lemma G.5.** *For all non-negative numbers $\alpha_1, \ldots \alpha_t \in [0, \alpha]$ with $\alpha_0 > 0$, we have:*

$$\sum_{t=1}^{T} \frac{\alpha_t}{\alpha_0 + \sum_{i=1}^{t-1} \alpha_i} \leq 2 + \frac{4\alpha}{\alpha_0} + 2\log\left(1 + \sum_{t=1}^{T-1} \frac{\alpha_t}{\alpha_0}\right).$$

*Proof.* Let us denote

$$T_0 = \min\{t \in [T] : \sum_{j=1}^{t-1} \alpha_j \geq \alpha\}.$$

Then, dividing the sum by $T_0$, we get:

$$\begin{aligned}
\sum_{t=1}^{T} \frac{\alpha_t}{\alpha_0 + \sum_{i=1}^{t-1} \alpha_i} &= \sum_{t=1}^{T_0-1} \frac{\alpha_t}{\alpha_0 + \sum_{i=1}^{t-1} \alpha_i} + \sum_{t=T_0}^{T} \frac{\alpha_t}{\alpha_0 + \sum_{i=1}^{t-1} \alpha_i} \\
&\leq \frac{1}{\alpha_0} \sum_{t=1}^{T_0-1} \alpha_t + \sum_{t=T_0}^{T} \frac{\alpha_t}{1/2\alpha_0 + 1/2\alpha + 1/2\sum_{i=1}^{t-1} \alpha_i} \\
&\leq \frac{\alpha}{\alpha_0} + 2\sum_{t=T_0}^{T} \frac{\alpha_i/\alpha_0}{1 + \sum_{i=T_0}^{t} \alpha_i/\alpha_0} \\
&\leq \frac{2\alpha}{\alpha_0} + 2 + 2\log\left(1 + \sum_{t=T_0}^{T} \alpha_i/\alpha_0\right) \\
&\leq \frac{2\alpha}{\alpha_0} + 2 + 2\log\left(1 + \sum_{t=1}^{T} \alpha_i/\alpha_0\right),
\end{aligned}$$

where we used the fact that $\sum_{j=1}^{T_0-1} \alpha_j \leq \alpha$ as well as for all $t \geq T_0$ we have that $\sum_{j=1}^{t-1} \alpha_j \geq \alpha$ (both follow from the definition of $T_0$) and [Lemma G.2](). $\qquad\square$

