# OpenReview forum: "Adaptive Bilevel Optimization"
_ICLR.cc/2024/Conference — Submitted to ICLR 2024_

### Official Review · Reviewer_1J7b · 2023-10-29

**Soundness:** 3 good
**Presentation:** 2 fair
**Contribution:** 3 good
**Rating:** 6
**Confidence:** 2

**Summary:**

The paper studies deterministic Bilevel optimization problems of the form
\begin{align*}
&\min_{x} f(x; y^\*(x)),\text{ where }y^\*(x)=\arg\min_{y}g(x;y)
\end{align*}
An algorithm is proposed which adapts to an unknown smoothness constant, assuming a known strong-convexity constant. The
algorithm guarantees $f(\bar x_T;y^\*(\bar x_T))-f(x^\*; y^\*(x^\*))\le O(1/T)$ in the convex bounded loss setting and $\min_t\\|\nabla f(x_t;y^\*(x_t))\\|\le O(1/T)$ in the non-convex unconstrained setting.

**Strengths:**

- The proposed algorithm does appear to adequately achieve adaptivity to the smoothness constant, as promised.
- The math is easy to follow and appears to be correct.
- The exposition is generally clear and free of typos. The experiment seems like a reasonable application and demonstration of the algorithm and includes several baselines for reference.

**Weaknesses:**

- The paper seems to focus on an easier deterministic optimization problem setting than its contemporaries such as Huang et al (2022), which develop similar algorithms for the stochastic Bilevel Optimization setting.
  In this sense the results may be rather limited in scope.
- The novelty of the algorithms is unclear to me: the approaches used seem similar to works such as Levy (2017) and Orabona (2023), which also show adaptivity to smoothness using normalized gradients. My main concern is that the results presented here are a straight-forward application of existing techniques to a new problem setting. This would be fine of course, but would of limited novelty.

Minor Gripes:
 - The experimental results would be easier to visually parse if you keep the same color mapping between experiments (e.g. ABO is black in one plot but green in another).
 - The conclusion mentions developing an adaptive step-size *schedule*, which feels like a contradiction of terminology: schedules are fixed in advance, like Cosine annealing or a linear decay schedule, whereas stepsize adaptation implies adapting the stepsize on-the-fly, as done in the algorithms presented here

References:
 - Huang, Feihu, et al. "Enhanced bilevel optimization via bregman distance." Advances in Neural Information Processing Systems 35 (2022): 28928-28939.
 - Levy, Kfir. "Online to offline conversions, universality and adaptive minibatch sizes." Advances in Neural Information Processing Systems 30 (2017).
 - Orabona, Francesco. "Normalized Gradients for All." arXiv preprint arXiv:2308.05621 (2023).

**Questions:**

- Could you elaborate on what new techniques were introduced that wouldn't be possible using existing results? Is there a reason that the convergence of the inner-loop, for instance, need to be re-proven and can't just call out to an existing result? The algorithm doesn't appear to be doing anything particularly surprising that I haven't encountered already, e.g. particularly in Levy (2017) or Orabona 2023

---

### Official Review · Reviewer_JR6w · 2023-10-30

**Soundness:** 2 fair
**Presentation:** 2 fair
**Contribution:** 2 fair
**Rating:** 3
**Confidence:** 4

**Summary:**

This paper studies and important problem of designing adaptive (parameter agnostic) algorithms for bilevel optimization. The main result of the work states that in deterministic setting, their algorithm converges with the rate O(1/T) in convex and non-convex cases. The key feature of the proposed algorithm is that it only requires the knowledge of the strong convexity parameter of the lower level problem and does not require smoothness constants. From technical side, the authors incorporate Adagrad-Norm type step-sizes in both inner and outer loop iterations.

**Strengths:**

1. The work proposed a nested adaptive algorithm and proved O(1/T) convergence rate in the deterministic setting without using the smoothness parameters of the problem.

2. Experiments are conducted to validate the efficiency of the proposed method.

**Weaknesses:**

1. Knowledge of strong convexity parameter $H_g$ is needed (and crucial in the analysis), which is a big weakness of the paper from theory side. In the deterministic setting (considered in the paper), this issue should not be too difficult to handle, e.g., using line-search or other adaptive methods. In minimization and min-max setting, such problem has been overcome even in the stochastic case, see, e.g., [1,2,3].

2. The authors do not make explicit the dependence on the constants and only report convergence in terms of $T$. This should be fixed before the paper gets accepted. It could happen that the dependence on problem parameters can be e.g., exponential, making it essentially useless to derive a polynomial dependence on $T$.

3. It is very confusing that Theorem 1 requires the bounded domain for convex case and unbounded domain (with Euclidean setting) for non-convex. A good theory should be able to handle both constraint/unconstrained, Euclidean/non-Euclidean cases at the same time.

4. There are several technical problems in the statements of theorems and some gaps in the proofs.

- Theorem 2 does not state any additional assumptions, while it becomes clear from Appendix F that a strong additional assumption (called reciprocity condition by authors) is needed. It seems that such assumption essentially means that the distance generating function $h(x)$ is smooth. This is very limiting for mirror descent framework.
- Page 21 $+\infty < +\infty$ is not necessarily a contradiction.
- Same page 21. In the proof, the diameter of $\mathcal X$ appears. However, it can be infinite in the unconstrained case. Thus, the next Theorem E.1. is wrong.

5. There a many incomplete, grammatically incorrect or repeating sentences in the draft. For example, in contributions section, first sentences in the first and third contributions missing a verb. In section 6, the sentence "It is known that..." is repeated.

6. In Assumption 2, what does it mean "relative to $y$"?

7. Use different markers in the plots, lines are indistinguishable when printed in gray-scale. What is the advantage of using Shannon entropy compared to the Euclidean distance?



[1] Yair Carmon, Oliver Hinder. Making SGD Parameter-Free. COLT 2022.

[2] J. Yang, X. Li, N. He. Nest your adaptive algorithm for parameter-agnostic nonconvex minimax optimization. NeurIPS 2022.

[3] X Li, J Yang, N He. TiAda: A Time-scale Adaptive Algorithm for Nonconvex Minimax Optimization. ICLR 2023.

**Questions:**

1. What is the motivation for using the Fenchel coupling in the step-sizes (12)? Why not using more common distance measures, e.g., symmetrized Bregman divergence, e.g., $D(x_k, x_{k+1}) + D(x_{k+1}, x_k)$. What is the connection with Fenchel coupling?

2. Is the set $\mathcal X$ in Definition 1 assumed to be closed? If yes, why consider its closure? If no, how to project?

3. I am curious why the authors decide to use Adagrad type step-sizes. Is it possible to simply use a line-search for this problem?

4. Why the update rule (7) is called "normalized". Usually the normalized gradient step has the property, usually normalization with a power $1$ is called Normalized GD.

5. On page 19, after "Focusing on the last term of the (RHS) we have:", why the second inequality holds?

Typos:
1. Lemma C.1. 2. $\partial h(x)$ should be $\partial h(x^+)$.
2. Lemma C.4. x^+ should be y^+.
3. On page 21, F(x^*, x_k) must be a typo, since x_k is in the primal space.
4. On page 23, "we will make the following ... inequality".

---

### Official Review · Reviewer_YefR · 2023-10-30

**Soundness:** 3 good
**Presentation:** 2 fair
**Contribution:** 2 fair
**Rating:** 5
**Confidence:** 4

**Summary:**

This work studied bilevel optimization with adaptive learning rate. Here adaptivity means that the developed algorithm is free from smoothness Lipschitz constants for both upper- and lower-level objective functions and learning rates. Compared to existing approaches that require heavily tuning for the theoretical guarantee and practical implementation, the proposed approach in this work can achieve similar convergence rate without such tuning efforts. The proposed method uses the idea from mirror descent. Motivated from Levy 2017, it uses a gradient norm accumulation idea to approximate the stepsizes. The outer-level stepsize is also designed to be adaptive based on an idea similarly to gradient mapping. In theory, they focus on the deterministic case, and obtain the same bounds as existing results but with less tuning efforts.

**Strengths:**

1.	This work studied a very interesting problem in bilevel optimization with less tuned parameters or without the requirement on knowledge of smoothness parameters. Given more complicated bilevel structure, getting less parameter tuning can be of interest in theory and in practice.

2.	The paper is well written. The story is clear, and the algorithm is easy to follow with each adaptive design carefully explained. Proof sketch is also provided for readers to check and understand the main idea.

3.	The idea of using the gradient mapping accumulation into the Fenchel coupling geometry to stabilize the convergence seems to be effective.

**Weaknesses:**

1.	The developed algorithm is not entirely adaptive. The stepsize $\eta_t$ requires the information of the strong-convexity parameter $H_g$ of the inner-level function $g$. This means that in experiments, you still need to tune the stepsizes. If I have a misunderstanding, please let me know.

2.	The algorithms are limited to the deterministic setting. Is this idea able to extend to stochastic case.

3.	The hypergradient $\tilde\nabla f$ requires an accurate computation of Hessian matrix inversion. So, a more practical case is to approximate the Hessian-inverse-vector product by solving a linear system or using the Neumann Series expansion. In this case, can the adaptive designs and analysis still work?

4.	There are some works on auto-tuning stepsizes for (stochastic) bilevel optimization. For example, [1] uses some idea from stochastic line search to auto-tune the learning rates without knowing the smoothness parameters. Their approach is also appliable in stochastic case. I feel that the proposed method should be also compared to bilevel methods with adaptive learning rates, which are missing in the comparison.

[1] Fan, Chen, Gaspard Choné-Ducasse, Mark Schmidt, and Christos Thrampoulidis. "BiSLS/SPS: Auto-tune Step Sizes for Stable Bi-level Optimization." arXiv preprint arXiv:2305.18666 (2023).

5.	In the experiments, the comparison algorithms (including the proposed one) work in the stochastic case with data sampling. For example, SABA, BSA, VRBO are all stochastic algorithms. However, the theory is given in the deterministic case. Thus, it is unclear whether the proposed adaptive designs should be further adjusted in the stochastic setting? This is important because the adaptive designs such as the introduced scaled norms in line 6, 8, 9 may be biased in the stochastic setting,

**Questions:**

Overall, I think this is an interesting and very important topic, and the authors provide a simple and encouraging method. However, given there are several concerns, I am on the slightly negative side, but I am ok to increase my score after the rebuttal.
See the weakness part for questions.

---

### Official Review · Reviewer_DLXB · 2023-10-31

**Soundness:** 2 fair
**Presentation:** 2 fair
**Contribution:** 2 fair
**Rating:** 3
**Confidence:** 5

**Summary:**

This paper proposes a new adaptive optimization algorithm based on mirror descent for a class of possibly nonconvex smooth bilevel problems with strongly-convex lower level. It provides the convergence analysis for the proposed algorithm and prove that it obtains a convergence rate $O(1/T)$.  Meanwhile, it provides some experimental results to verify the efficiency of the proposed algorithm.

**Strengths:**

This paper proposes a new adaptive optimization algorithm based on mirror descent for a class of possibly nonconvex smooth bilevel problems with strongly-convex lower level. It provides the convergence analysis for the proposed algorithm and prove that it obtains a convergence rate $O(1/T)$.  Meanwhile, it provides some experimental results to verify the efficiency of the proposed algorithm.

**Weaknesses:**

Although this paper assert that the proposed adaptive algorithm does not rely on the Lipschitz constants, it strictly depends on the unknown strong convexity parameter $H_g$ of function $g(x,\cdot)$ and strong convexity parameter $K$ of function $h(\cdot)$. Meanwhile, it requires to compute Hessian matrix and its inverse. This algorithm is a double-loop algorithm, and number of iteration in inner loop increases as number of iteration in outer loop increases. The proposed algorithm only considers the deterministic bilevel problems but does not consider the stochastic problems. Clearly, the proposed algorithm can not be efficiency in solving large-scale problems. Recently, there exist many efficient single-loop Hessian-free or full first-order gradient algorithms.  Meanwhile, the convergence analysis of the proposed algorithm basically follows the existing convergence analysis. In summary, the novelty of  this paper is limited. Thus, the level of this paper does not reach the level of ICLR.

**Questions:**

1)	In the proposed algorithm, the function $h(\cdot)$ is regularizer. For easily reading, please give a specific example for function $h(\cdot)$.

2)	In the experiments, how to choose this regularizer $h(\cdot)$ ?

3)	In the proposed algorithm, given $G_t = G_{t-1}+|| \nabla _y g(x_k,y_t)||$. If $|| \nabla _y g(x_k,y_t)|| $ is not bounded, the proposed algorithm is meaningless. Thus, the algorithm implicitly assume that $ || \nabla _y g(x_k,y_t)|| $ is bounded, which is a stricter assumption than the existing bilevel algorithms.

4)	In the convergence analysis, the authors only provide the convergence results of the proposed algorithm on the convex function $f$ and the bounded $\mathcal{X}$. When the function $f$ is convex and the set $\mathcal{X}$ is not bounded, the authors can also provide its convergence results?

5)	In the convergence analysis, the authors only provide the convergence results of the proposed algorithm on the non-convex function $f$ and $\mathcal{X}=\mathbb{R}^m$. When the function $f$ is non-convex and the set $\mathcal{X} \subset\mathbb{R}^m $, the authors can also provide its convergence results?

---

### Official Review · Reviewer_KS49 · 2023-10-31

**Soundness:** 3 good
**Presentation:** 3 good
**Contribution:** 2 fair
**Rating:** 6
**Confidence:** 3

**Summary:**

This paper proposed an adaptive algorithm for solving bilevel optimization problems with strongly convex inner function. If outer function is convex, the algorithm achieves a convergence rate of $\mathcal{O}(1/T)$ in terms of the outer objective function. If the outer objective is non-convex, the algorithm achieves an $\mathcal{O}(1/T)$ best-iterate guarantee for the squared norm of the gradient of the outer objective function.

**Strengths:**

Originality: The authors proposed a adaptive optimization algorithm based on mirror descent for solving bilevel optimization problem without knowing the Lipschitz constants.

Quality: Compared with approximation bilevel method, their approach is more robust and practical based on the numerical experiments.

Clarity: The overall structure and presentation of the paper is clear and easy to follow.

**Weaknesses:**

1. The authors stated it is the first adaptive method for solving bilevel optimization problems. But I saw a paper [a] about adaptive methods for stochastic bilevel optimization problem. The statement is not true.

2. Although the algorithm utilized the adaptive method without knowing the Lipschitz constants, it did not achieve the state-of-the-art convergence rate. The state-of-the-art convergence rate for convex-strongly-convex case is $\tilde{\mathcal{O}}(1/T^2)$ in [b]. Is it possible to improve the proposed algorithm to that rate?

3. All the numerical results are plotted in terms of iterations. It would be better if the authors provide some results in terms of running time.

References:

[a]. BiAdam: Fast Adaptive Bilevel Optimization Methods . Feihu Huang, Junyi Li, Shangqian Gao.

[b]. Lower Bounds and Accelerated Algorithms for Bilevel Optimization. Kaiyi Ji, Yingbin Liang.

**Questions:**

All my questions are listed in the Weaknesses section.

---

### Meta-Review · Area_Chair_taMY · 2023-12-06

**Metareview:**

There are many issue raised by the review team, but the authors did not response to any review's comment.

**Justification For Why Not Higher Score:**

There are many issue raised by the review team, but the authors did not response to any review's comment.

**Justification For Why Not Lower Score:**

N/A

---

### Decision · Program_Chairs · 2024-01-16

Reject